# Extracellular Vesicles: Advanced Tools for Disease Diagnosis, Monitoring, and Therapies

**DOI:** 10.3390/ijms26010189

**Published:** 2024-12-29

**Authors:** Pedro Lorite, Jorge N. Domínguez, Teresa Palomeque, María Isabel Torres

**Affiliations:** Department of Experimental Biology, Faculty of Health Sciences, University of Jaén, 23071 Jaén, Spain; plorite@ujaen.es (P.L.); jorgendm@ujaen.es (J.N.D.); tpalome@ujaen.es (T.P.)

**Keywords:** exosomes, microvesicles, extracellular vesicles, apoptotic bodies, cargo packaging, EV therapies, drug delivery, EV biogenesis

## Abstract

Extracellular vesicles (EVs) are a heterogeneous group of membrane-encapsulated vesicles released by cells into the extracellular space. They play a crucial role in intercellular communication by transporting bioactive molecules such as proteins, lipids, and nucleic acids. EVs can be detected in body fluids, including blood plasma, urine, saliva, amniotic fluid, breast milk, and pleural ascites. The complexity and diversity of EVs require a robust and standardized approach. By adhering to standardized protocols and guidelines, researchers can ensure the consistency, purity, and reproducibility of isolated EVs, facilitating their use in diagnostics, therapies, and research. Exosomes and microvesicles represent an exciting frontier in modern medicine, with significant potential to transform the diagnosis and treatment of various diseases with an important role in personalized medicine and precision therapy. The primary objective of this review is to provide an updated analysis of the significance of EVs by highlighting their mechanisms of action and exploring their applications in the diagnosis and treatment of various diseases. Additionally, the review addresses the existing limitations and future potential of EVs, offering practical recommendations to resolve current challenges and enhance their viability for clinical use. This comprehensive approach aims to bridge the gap between EV research and its practical application in healthcare.

## 1. Introduction

Extracellular vesicles (EVs) represent a diverse group of membranous structures released by cells into the extracellular space. These vesicles, which include exosomes, microvesicles, and apoptotic bodies, among others, have emerged as key components in intercellular communication and the transport of biomolecules. Their ability to shuttle proteins, lipids, and nucleic acids between cells underscores their significance in numerous biological and pathological processes [1,2,3].

EVs play an essential role in regulating various cellular functions and physiological processes, such as immune modulation, response to environmental stimuli, and maintenance of cellular homeostasis [4,5]. Moreover, they are involved in disease progression, including cancer, neurodegenerative diseases, and cardiovascular disorders, where they act both as mediators of disease and potential indicators of its presence and evolution [6,7,8,9,10].

The complexity and heterogeneity of EVs necessitate a meticulous approach to their isolation and characterization [11,12]. Current methods, including techniques such as ultracentrifugation, filtration, and chromatography, must be rigorously standardized to ensure the purity and reproducibility of the isolated EVs [13,14,15]. These efforts are crucial for leveraging the diagnostic and therapeutic potential of EVs. With advances in our understanding of the biogenesis, function, and molecular profiles of EVs, these vesicles are emerging as valuable tools in personalized medicine and precision therapy [16,17].

The intracellular trafficking of EVs is influenced by a range of factors, including physicochemical properties, cell types, environmental conditions, and stimulatory agents. Different cell types synthesize and release distinct EVs with unique compositions and functions, while cell state—such as active proliferation, quiescence, or stress—also modulates EV production and release. Environmental conditions, including temperature, pH, and ion concentration, further affect EV stability, activity, and interactions with surrounding cells [18,19]. Stimulatory factors, such as Ca^2+^ ionophores, hypoxia, and cell detachment, can trigger EV secretion, influencing their intracellular transport and function [20]. In cell culture settings, the presence or absence of serum impacts cell monolayer formation and cellular integrity, which in turn can alter EV biogenesis. Each phase of the EV lifecycle—including membrane invagination, vesicle formation, cargo loading, and secretion—is sensitive to these factors, which collectively shape the processes underlying EV intracellular transport [19].

The exploration of extracellular vesicles offers valuable insights into cellular functions and disease mechanisms while presenting new opportunities for the development of advanced diagnostic and therapeutic strategies. As research continues to elucidate the multifaceted roles of EVs, their incorporation into clinical practice has the potential to transform approaches to disease diagnosis and treatment, providing more targeted and effective solutions.

In conclusion, this review stands out by integrating technological advancements, omics approaches, and emerging clinical applications, offering a current and comprehensive resource that can serve as a key resource for researchers and clinicians interested in translating EVs from the laboratory to practical medical applications.

## 2. EVs Classification

EVs are generally classified based on their size, biogenesis, and content [21,22]. The classification of extracellular vesicles includes exosomes, microvesicles, apoptotic bodies, and emerging categories such as large oncosomes, ectosomes, migrasomes, exomeres, and arrestin domain-containing protein 1-mediated microvesicles (Figure 1).

***Exosomes*** (30–150 nm) are formed within the endosomal system through the invagination of endosomal membranes, creating multivesicular bodies (MVBs) containing intraluminal vesicles [23,24]. When MVBs fuse with the plasma membrane, exosomes are released into the extracellular space (Figure 2). They are characterized by specific markers that aid in their isolation, characterization, and functional studies. Commonly used markers include tetraspanins (CD9, CD63, CD81), endosomal sorting complex proteins (TSG101, Alix), heat shock proteins (HSP70, HSP90), and membrane-associated proteins (Flotillin-1 and Flotillin-2) [25,26,27]. Exosomes contain a variety of bioactive molecules, including proteins, lipids, RNA (mRNA, miRNA), and DNA, reflecting the physiological or pathological state of the originating cell (Table 1). Exosomes are involved in intercellular communication, immune responses, and the transfer of genetic material, playing roles in various physiological processes and diseases, including cancer, neurodegenerative diseases, and cardiovascular conditions [24,28].

***Microvesicles*** (100–1000 nm) are formed by the outward budding and fission of the plasma membrane, involving cytoskeletal reorganization and the shedding of vesicles directly from the cell surface [29] (Figure 2). They carry proteins, lipids, and nucleic acids, typically derived from the plasma membrane and cytoplasm of the parent cell [16]. Like exosomes, microvesicles participate in intercellular communication and can influence various physiological and pathological processes, such as coagulation, inflammation, and cancer progression [30,31]. Markers used for the isolation, characterization, and functional studies of microvesicles include plasma membrane proteins (flotillins, annexin V, integrins like CD41, CD61, selectins like CD62, phosphatidylserine detected by annexin V binding), specific cell surface markers for endothelial cells (CD31, CD144), platelets (CD41, CD61, CD42b), leukocytes (CD45, CD11a, CD11b), erythrocytes (glycophorin A), tumor cells (EpCAM, HER2/neu), and MHC class I and II molecules [29,32] (Table 1).

Exosomal content includes proteins from endosomes, the plasma membrane, the cytosol, and specific subsets of cellular proteins depending on the cell type, whereas microvesicles can contain a variety of molecular cargos [21]. Although significant progress has been made in understanding the contents of both types of vesicles, identifying specific cargos remains an active area of research. Both microvesicles and exosomes can encapsulate and transfer proteins, RNA transcripts, and miRNA, but recent findings indicate differences in their capacity to transfer molecules from transiently transfected cells. While both types can efficiently incorporate exogenous proteins and mRNA, only microvesicles can transfer reporter functions to recipient cells in the form of plasmid DNA [3].

***Apoptotic Bodies*** (1000–5000 nm) are released during programmed cell death (apoptosis). As cells undergo apoptosis, they form membrane blebs that eventually detach to form apoptotic bodies [33,34]. These bodies contain fragmented DNA, histones, and other nuclear material, as well as cytoplasmic organelles and cell debris. Apoptotic bodies play a role in the clearance of dying cells by phagocytic cells, thereby preventing the release of potentially harmful cellular contents into the extracellular space. They may also participate in intercellular signaling and the transfer of cellular components. Apoptotic bodies contain cellular debris, nuclear fragments, and intact organelles. Common markers for the isolation, characterization, and functional analysis of apoptotic bodies include plasma membrane markers (externalized phosphatidylserine detected by Annexin V binding), nuclear markers (histones indicating the presence of nuclear material), and cytoplasmic markers (cytokeratins specific to the cell type undergoing apoptosis and actin-associated with the cytoskeleton) [35] (Table 1).

Although a definitive categorization has not yet been achieved, EVs can be broadly classified into three main categories based on their biogenesis and markers: (i) ectosomes (also known as shedding microvesicles), (ii) exosomes, and (iii) apoptotic bodies.

Advances in cellular and molecular biology have revealed a more complex landscape of EVs, leading to the identification of several emerging categories that exhibit unique characteristics and functions. These emerging categories include migrasomes, large oncosomes, exomeres, ectosomes, and arrestin domain-containing protein 1-mediated microvesicles (ARMMs). These other EV types, each with unique biogenesis routes and/or chemical features, yet these subtypes are difficult to separate due to overlapping characteristics [36,37,38,39,40,41,42].

***Large Oncosomes*** are a subtype of microvesicles with a size range of 1–10 μm, released by tumor cells. They form through budding of the plasma membrane and are frequently associated with highly aggressive tumors. These vesicles contain significant amounts of proteins, lipids, and nucleic acids, reflecting the transformed state of the tumor cells. Large oncosomes are implicated in the propagation of oncogenic signals and the modulation of the tumor microenvironment, promoting cancer progression and metastasis [36].

***Ectosomes*** are a type of extracellular vesicle released from the plasma membrane of cells, triggered by various stimuli such as cellular stress, activation, or apoptosis. Their formation involves cytoskeletal rearrangements and plasma membrane budding, followed by scission to release the vesicle into the extracellular space. Ectosomes are typically larger than exosomes, ranging from 100 nm to 1000 nm in diameter. They have a heterogeneous, often spherical shape and are surrounded by a lipid bilayer derived from the plasma membrane. This lipid composition includes phosphatidylserine, which is often externalized on the surface of ectosomes [37]. Ectosomes contain a variety of proteins, including membrane proteins (e.g., integrins, selectins), cytoskeletal proteins (e.g., actin, myosin), and enzymes. They can transport mRNA, microRNAs (miRNAs), and other non-coding RNAs. Ectosomes may also contain fragments of genomic and mitochondrial DNA [37]. They play a role in cell-to-cell communication by transferring bioactive molecules between cells. Ectosomes can modulate immune responses, for instance, by carrying antigens or signaling molecules. Platelet-derived and other cell-derived ectosomes may participate in the coagulation cascade. They are involved in pathological processes, including cancer progression, inflammation, and cardiovascular diseases.

***Exomeres*** are a recently discovered class of non-membranous nanoparticles, distinct from classical EVs, with a size of <50 nm. The exact biogenesis pathway of exomeres is still under investigation. Exomeres carry a specific set of proteins and nucleic acids with a composition that differs significantly from exosomes and microvesicles. While their precise biological roles are not yet fully understood, exomeres are believed to be involved in metabolic regulation and cellular communication [38].

***Migrasomes*** category of EVs that play a significant role in cell communication, particularly during cell migration. These vesicles are characterized by their unique origin and structure, migrasomes form during a specific process known as migracytosis, which occurs when cells undergo migration. As cells move, they extend structures called retraction fibers, which are thin extensions of the plasma membrane [39]. This detachment process encapsulates portions of the cytoplasm within the migrasomes, along with various proteins, lipids, and other cellular components. They are typically larger than other EVs, with sizes ranging from 500 nanometers to several micrometers in diameter. They possess a complex internal structure, often appearing as large, balloon-like vesicles containing multiple smaller vesicular structures, a feature that distinguishes them from the more uniformly sized and shaped exosomes and microvesicles. The primary role of migrasomes is thought to be in intercellular communication, particularly in the context of tissue development and repair. By packaging and releasing cellular contents during migration, migrasomes can influence the behavior of nearby cells, potentially by delivering signaling molecules or enzymes that modulate the local environment. Migrasomes are also implicated in the removal of cellular debris and damaged components, a process important for maintaining tissue homeostasis. Their formation during cell migration suggests they might play roles in processes like embryogenesis, wound healing, and cancer metastasis, where cell migration is a critical component [40].

***Arrestin Domain-Containing Protein 1-Mediated Microvesicles***, commonly referred to as ARMMs, represent a specialized subcategory of extracellular vesicles. ARMMs are formed through a distinct budding process from the plasma membrane of cells. This budding mechanism is similar to the viral budding process and is facilitated by arrestin domain-containing protein 1 (ARRDC1), which interacts with the endosomal sorting complexes required for transport (ESCRT) machinery and is directly shed from the cell surface. The biological functions of ARMMs are still being elucidated, but they are believed to play significant roles in cell signaling and communication. Similar to other EVs, ARMMs can carry a variety of bioactive molecules, including proteins, lipids, and nucleic acids [41,42]. Given their unique formation process, ARMMs are hypothesized to be involved in specific cellular functions that require the direct release of vesicles from the plasma membrane. This may include roles in immune responses, cancer progression, and other physiological or pathological processes where rapid and direct intercellular communication is advantageous [41].

The 2023 International Society for Extracellular Vesicles (ISEV) guideline emphasizes the importance of standardizing the terminology used for EVs [43]. Given the growing interest in EV research, it has become crucial to ensure consistency in naming conventions, classifications, and descriptions to facilitate clear communication across the scientific community. The guidelines recommend using “EVs” as a general term to describe all types of vesicles released from cells while avoiding assumptions about their specific biogenesis or function unless there is direct evidence. Specific categories of EVs, such as exosomes and microvesicles, should only be used when their origin or size has been confirmed by validated methods. Additionally, the guidelines suggest avoiding potentially misleading terms and encourage the use of quantitative descriptors, such as size, density, and cargo content, to provide a clearer understanding of EV populations [43]. This standardization will improve reproducibility and reliability in research, ensuring a common language for scientists in the field.

## 3. EVs Cargo

EVs comprise a heterogeneous population whose function is ultimately determined by the vesicular cargo content, which is reflective of their cell of origin and the physiological or pathological state of the cell. The macromolecular cargo contained within microvesicles participates in a wide range of biological processes [44].

The main types of cargo found in EVs are proteins, nucleic acids, lipids, metabolites, and others. In this sense, they include proteins from the plasma membrane or endosomal membranes, such as tetraspanins (CD9, CD63, CD81), integrins, and MHC molecules [18]. Also, cytosolic proteins, enzymes, signaling molecules, and cytoskeletal proteins like actin and tubulin. Thus, extracellular matrix proteins, including fibronectin, collagens, and laminins. Heat Shock Proteins (HSPs) HSP70 and HSP90 are commonly found in EVs, playing roles in protein folding and stress responses [45].

The nucleic acids messenger RNA (mRNA), which can be translated into proteins in recipient cells. MicroRNA (miRNA) can regulate gene expression post-transcriptionally in target cells. Other Non-Coding RNAs include long non-coding RNAs (lncRNAs), small nuclear RNAs (snRNAs), and circular RNAs (circRNAs). Some EVs can carry double-stranded or single-stranded DNA fragments, including mitochondrial DNA [46].

EVs possess membranes rich in phospholipids, cholesterol, and sphingolipids, with a lipid composition that often mirrors that of the parent cell’s membrane. EVs also transport lipid signaling molecules, such as prostaglandins, sphingosine-1-phosphate, and other bioactive lipids, playing a key role in intercellular communication and various biological processes [47].

EVs can contain small metabolites, including amino acids, sugars, and other small molecules involved in metabolism. EVs can carry drugs, nanoparticles, or other therapeutic agents if cells are engineered or exposed to these substances. In some cases, EVs can contain viral particles or bacterial components, aiding in pathogen dissemination or immune modulation [48].

With the induced interest in EVs, the amount of data generated has increased exponentially. There are several databases as a crucial resource for researchers in the field of extracellular vesicle research, offering data that can be used for discovering biomarkers, understanding cellular communication, and exploring therapeutic potentials. Researchers often use in conjunction these bioinformatics tools to analyze the functional implications of vesicle components [49,50,51,52] (Table 2). The EV-TRACK knowledgebase centralizes (meta) data of EV separation and characterization where researchers can share protocols, promote methodological rigor, and ensure data reproducibility [53]. By standardizing the reporting of EV isolation and characterization techniques, EV-TRACK facilitates reliable comparisons across studies and accelerates clinical translation of EV-based technologies.

## 4. EVs Mechanisms of Action and Integrative Proteo-Transcriptomic Analyses

### 4.1. EVs Mechanisms of Action

EVs play pivotal roles in disease by acting as carriers of bioactive molecules, which modulate several processes in the recipient cells. There is a common function associated with the cargo content of extracellular vesicles based on the types of molecules found within them:

#### 4.1.1. Cellular Communication

EVs play a pivotal role in cellular communication by transporting bioactive molecules between cells, allowing them to influence the physiological and pathological states of recipient cells, acting as messengers that facilitate various types of intercellular communication, contributing to both health and disease [54]. The action mechanism of EVs in cellular communication involves several key steps: EV biogenesis and release enriched with specific bioactive molecules from the parent cell [4]; target cell recognition and uptake through interactions between specific surface proteins, lipids, or receptors to guide the EVs to their appropriate destination; cargo delivery mechanisms to recipient cells via several mechanisms as direct fusion, endocytosis, and receptor-mediated signaling; functional impact on recipient cells, include gene regulation, protein transfer and metabolic regulation; physiological and pathological roles, such as immune regulation, tissue repair, and development. [30,55,56].

One of the primary mechanisms by which EVs communicate with target cells is through endocytosis. EVs are internalized by recipient cells via clathrin-mediated endocytosis, caveolin-dependent pathways, or macropinocytosis. Once inside the cell, the vesicle’s cargo is released into the cytosol, where it can exert its effects on cellular processes. Another mechanism is the fusion with the plasma membrane of the recipient cell, releasing their content directly into the cytoplasm, avoiding endosomal sequestration [54,55]. EVs facilitate long-distance communication by traveling through biological fluids, such as blood, urine, or saliva, to distant tissues. This allows for systemic signaling and coordination of responses, such as in the context of tissue repair or tumor metastasis [57,58]. EVs influence cellular phenotype in recipient cells, such as promoting proliferation, differentiation, or migration.

#### 4.1.2. Immunomodulation and Inflammation

EVs contain proteins and other factors that can influence immune responses, acting in both the activation and suppression of the immune system [59,60]. By transferring bioactive molecules, EVs modulate immune functions such as antigen presentation, inflammation, and immune suppression. EVs, particularly exosomes derived from antigen-presenting cells, play a role in the immune response by presenting antigens to T cells. These exosomes contain major histocompatibility complex molecules, both class I and class II, as well as co-stimulatory molecules, enabling them to effectively trigger T cell responses, enhancing their ability to activate CD4+ and CD8+ T cells [60]. Extracellular vesicles can also contribute to immune suppression, carrying immuno-suppressive molecules such as PD-L1, TGF-β, and Fas ligand, which can inhibit the activity of CTLs and NK cells. Macrophages and neutrophils release EVs that can carry pro-inflammatory or anti-inflammatory signals, depending on the context [61]. One notable example is EVs carrying annexin A1, an anti-inflammatory protein that modulates immune cell recruitment and reduces the production of pro-inflammatory cytokines [62].

#### 4.1.3. Genetic Material Transport

EVs can transfer mRNA and miRNA to recipient cells, affecting gene expression and contributing to processes such as proliferation, differentiation, and apoptosis [63]. EVs can transfer DNA, particularly fragmented genomic DNA and mitochondrial DNA. This process, often referred to as horizontal gene transfer, allows for the exchange of genetic information between cells in a manner that could lead to genetic reprogramming in recipient cells [64]. These molecules can modulate gene expression in recipient cells by altering their transcriptional and translational machinery. Importantly, EVs act as a means of transferring genetic information in a manner that is protected from degradation by extracellular nucleases, thanks to the vesicular lipid bilayer. Through the encapsulation and delivery of genetic material, EVs enable dynamic changes in recipient cells, influencing a variety of physiological and pathological processes. As our understanding of their mechanisms continues to grow, EVs represent a promising frontier in the treatment of diseases and gene therapy applications.

#### 4.1.4. Tumor Metastasis and Angiogenesis

The action mechanism of EVs in tumor metastasis involves several key steps that help cancer cells spread, survive, and thrive at secondary sites. These steps include the formation and release of EVs by tumor cells, their interaction with recipient cells, and the functional changes they induce to promote metastasis [65]. Tumor cells produce and release large quantities of EVs into the tumor microenvironment and bloodstream. EVs released from primary tumors travel to distant organs and modify the local environment to create a “pre-metastatic niche” that is favorable for incoming tumor cells [66]. This includes changes in extracellular matrix (ECM) composition, inflammation, immune cell recruitment, and angiogenesis (new blood vessel formation) to support tumor growth. EVs carry pro-angiogenic factors, such as vascular endothelial growth factor (VEGF) and basic fibroblast growth factor (bFGF), as well as miRNAs like miR-126 and miR-210, which have been implicated in enhancing endothelial cell function and angiogenesis [67]. They also promote the formation of new blood vessels that supply nutrients and oxygen to growing tumors and provide a route for cancer cells to enter the bloodstream and disseminate to other parts of the body. EVs can also induce epithelial-to-mesenchymal transition (EMT) in cancer cells, leading to the loss of epithelial characteristics, including cell adhesion, and the gain of mesenchymal traits, such as increased mobility and invasiveness [68]. Additionally, EVs carry matrix-degrading enzymes, such as matrix metalloproteinases (MMPs), which break down the ECM, creating space for tumor cells to invade and establish new metastatic colonies. At secondary sites, EVs can enhance the stem cell-like characteristics of cancer cells, giving them greater adaptability and resistance to therapies [69].

#### 4.1.5. Cellular Homeostasis

Cell protection and repair processes. EVs play a crucial role in protecting cells by mediating various mechanisms, including stress response, immune modulation, and the preservation of cellular integrity [70]. One of the primary protective mechanisms of EVs is their involvement in the cellular stress response. Under conditions such as oxidative stress, hypoxia, or inflammation, cells release EVs loaded with stress-related molecules, including HSPs and anti-oxidants [71]. EVs also remove toxic substances from cells, acting as a detoxification mechanism. Cells can package harmful materials, such as misfolded proteins, into EVs and export them out of the cell, thus maintaining cellular health. Another important protective function of EVs is their role in apoptosis regulation. EVs can convey anti-apoptotic signals to cells at risk of undergoing programmed cell death. This mechanism is particularly relevant in the context of tissue injury and regeneration, where EVs help prevent excessive cell loss, ensuring tissue integrity and repair [33].

EVs from senescent cells have complex roles, impacting tissue health and disease by altering surrounding cells’ behavior contributing to processes such as aging, tissue regeneration, and cancer progression. EVs from senescent cells can transfer senescence-inducing signals to surrounding cells, promoting a “bystander” effect where nearby cells also enter a state of senescence. The EVs can either recruit immune cells to clear the senescent cells or suppress immune activity, depending on the context, affecting inflammation and tissue homeostasis [72,73].

#### 4.1.6. Modulation of Cellular Metabolism

EVs play a significant role in the modulation of cellular metabolism, influencing a variety of metabolic processes in recipient cells. Through the transfer of bioactive molecules, EVs can alter metabolic pathways, impacting cellular energy production, nutrient utilization, and metabolic homeostasis [74]. EVs can modulate cellular metabolism by transferring enzymes and proteins involved in metabolic processes. For example, they may carry glycolytic enzymes, which can enhance or modulate glucose metabolism in recipient cells. EVs can transfer lipids like ceramides and sphingomyelin, which are involved in regulating cellular processes like apoptosis, proliferation, and stress responses [75]. In adipose tissue, EVs can influence lipid storage and mobilization, contributing to the metabolic regulation of fat cells and potentially impacting systemic metabolism [76]. EVs have been found to transfer mitochondrial components and even intact mitochondria to recipient cells, thereby affecting their mitochondrial function and energy production [77]. This transfer can influence the bioenergetic status of cells by enhancing or restoring mitochondrial activity. EVs can also modulate nutrient-sensing pathways, such as the mTOR and AMPK signaling pathways, which are central regulators of cellular metabolism and growth [78].

Due to their role in disease and health, EVs are being studied for their potential as biomarkers for disease diagnosis and prognosis, as well as for their therapeutic potential in delivering drugs and other treatments to specific cells [7,79].

### 4.2. Integrative Proteo-Transcriptomic Analyses

The integration of proteomic and transcriptomic data has unveiled that extracellular vesicles are not merely passive carriers of biomolecules but active sites of intricate RNA-protein interactions. These RNA-protein complexes (RNPs) form functional networks within EVs, playing crucial roles in regulating RNA stability, accessibility, and activity. Integrative proteo-transcriptomic analyses have emerged as a powerful platform for decoding the mechanisms underlying EV-mediated functions, offering a comprehensive view of the interplay between EV proteins and RNAs [80].

Currently, most EV research focuses on either proteomic or transcriptomic signatures independently [81,82]. While extensive studies have characterized donor cells and the content of their EVs, the interconnected functionalities between EV proteins and RNAs remain underexplored [83]. This gap largely stems from the limited availability of studies combining proteomic and transcriptomic analyses of matching donor cells and their EVs. By addressing this shortfall, integrative proteo-transcriptomic analyses of EVs can reveal the mutual regulation between RNA and proteins, providing essential insights for applications such as advanced therapeutic strategies and non-invasive next-generation liquid biopsy techniques [84].

Such integrative approaches simultaneously assess the proteomic and transcriptomic content of EVs, uncovering the co-packaging of RNA and proteins and their potential synergistic roles. For example, RNA-binding proteins such as hnRNPA2B1, YBX1, and AGO2 are frequently identified in EVs and play essential roles in miRNA sorting, stability, and function, interacting with specific RNA motifs, mediating the selective packaging of RNAs into EVs [85]. RNPs protect RNAs from degradation, regulate their activity, and enable targeted delivery to recipient cells, where they influence gene expression and cellular functions. By mapping the composition and organization of RNPs, researchers can better understand their roles in EV-mediated communication.

The implications of studying RNA-protein complexes in EVs are profound, particularly for unraveling disease mechanisms. Dysregulation of EV-associated RNPs has been linked to pathological processes, including tumor progression, immune evasion, and the dissemination of infectious agents [86,87]. Moreover, identifying specific RNP signatures in EVs holds promise for the development of novel biomarkers for early disease detection and monitoring. These insights exemplify the convergence of fundamental molecular biology and translational research, emphasizing the potential of EV-based diagnostics and therapeutics.

## 5. Isolation Methods, Quantification, and Purity Assessment of EVs

This review provides an overview of current EV isolation techniques, addressing their challenges and limitations, and highlights potential pathways for achieving EV isolation. The isolation, quantification, and purity assessment of EVs are foundational to EV research. The choice of methods depends on research goals, sample characteristics, and available resources. Optimizing and validating these processes are crucial for advancing EV research, ensuring accurate results, and supporting translational applications [88,89].

The significant heterogeneity of EVs, their low abundance in biofluids, and technical inconsistencies in sample collection present major challenges for EV isolation. Current strategies to isolate EVs include differential centrifugation (DC) [90], filtration using hydrophilic polyvinylidene difluoride (PVDF) membranes of different pore sizes [91], high-performance size-exclusion chromatography (SEC) [92], ultrafiltration with SEC, immunocapture [82], heparin affinity purification [93], differential density-gradient ultracentrifugation, tangential flow filtration [94], field-flow fractionation, synthetic polymer-based precipitation [95], and microfluidic isolation [96]. While each method has its limitations, these advancements have been crucial in driving EV research forward. Gaining a deeper understanding of the strengths and limitations of each technique can enable researchers to optimize methods for isolating specific EV subtypes.

### 5.1. Pre-Isolation Considerations

Sample collection and handling. Standardize procedures for the collection, storage, and handling of biological fluids, including consistent use of anticoagulants, processing times, and storage conditions to prevent degradation or alteration of EVs. Implement standardized protocols for initial sample processing, such as centrifugation steps to remove cells, debris, and larger particles [89].

### 5.2. Common Isolation Methods

Extracellular vesicles play pivotal roles in both physiological and pathological conditions. However, a major challenge in the field of EVs is their heterogeneity and the methods used to isolate and purify distinct populations. Several methods are employed, each with distinct principles, advantages, and limitations (Figure 3, Table 3).

Differential ultracentrifugation is widely used for isolating EVs by sequential centrifugation at increasing speeds, which separates particles based on size and density. Low-speed centrifugation removes cells and debris, intermediate-speed centrifugation eliminates organelles and larger particles, and finally, high-speed ultracentrifugation (~100,000× *g*) isolates EVs. It is widely used and standardized in EV research because there is no need for additional reagents, minimizing chemical contamination, and it is relatively cost-effective for labs with ultracentrifuge access. However, it has some limitations, as it may co-isolate protein aggregates and non-vesicular particles, requiring expensive equipment and maintenance. The prolonged high-speed centrifugation can damage EV integrity, which is labor-intensive and not easily scalable for large sample volumes [97].Ultrafiltration uses membranes with defined pore sizes to filter particles, retaining EVs within a specific size range (30–200 nm). This method presents advantages as it is faster than ultracentrifugation with no specialized equipment required, and it is more affordable, preserving EV structure better than ultracentrifugation. Also, it presents limitations as membranes can clog with proteins or debris, reducing efficiency. It presents a limited ability to handle large volumes and has a risk of contamination from non-vesicular particles adhering to membranes [98].Polymer-based precipitation polymers like polyethylene glycol (PEG) induce precipitation of EVs and other macromolecules from the solution. This method is simple and accessible, with no specialized equipment needed. It is suitable for processing large sample volumes and is time-efficient. It presents low purity because it co-precipitates with proteins and other impurities, requiring additional purification steps for downstream applications. The reproducibility can be inconsistent due to variability in polymer performance [99].Size-exclusion chromatography (SEC) particles are separated based on size as they pass through a column packed with porous beads. Larger EVs elute earlier, while smaller particles and proteins are retained longer. Present high purity effectively separates EVs from contaminants. It preserves EV integrity and functionality, and it is compatible with automated systems for reproducibility. It has lower throughput compared to other methods requiring high-quality columns, which can be expensive and challenging to scale for industrial or clinical applications [100].Immunoaffinity-based isolation uses antibodies targeting EV surface markers (e.g., CD63, CD81, CD9) immobilized on magnetic beads or other matrices with high specificity that enables isolation of specific EV subpopulations. It is ideal for detailed molecular or functional studies but has a high cost of antibodies and materials with limited scalability for large volumes and potential contamination with residual antibodies or reagents [101].Microfluidic technologies. Isolating extracellular vesicles is challenging due to the variability of biological fluids in composition, viscosity, and contamination. Specific issues include protein contaminants in plasma and serum, high viscosity in fluids like synovial fluid or semen, and limited sample volumes in fluids like cerebrospinal fluid (CSF) or amniotic fluid, demanding efficient isolation methods that minimize sample loss while maintaining EV integrity. Microfluidics-based technologies are at the forefront of addressing these challenges, offering scalable, sensitive, and precise solutions. Two primary principles guide their application: size-based separation and immunocapture. Size-based separation isolates EVs by exploiting size differences, using methods like filtration, hydrodynamic flow, and dielectrophoresis [102]. These techniques ensure high throughput, gentle handling, and adaptability across biofluids, especially for plasma. However, limitations include overlapping size ranges with contaminants, clogging risks, and reduced specificity for EV subtypes, requiring optimization to enhance purity and reliability. Magnetic Bead-Based Immunocapture combines microfluidic systems with antibody-conjugated magnetic beads to selectively bind EVs. Aptamer-based systems use nucleic acid aptamers as ligands to bind specific EV markers, providing high specificity and versatility. Microfluidic chips functionalized with antibodies targeting specific surface markers allow for selective capture of EV subpopulations. This approach is beneficial for isolating tumor-derived EVs from blood or specific EV populations from urine. Integrated microfluidic systems now combine preprocessing steps, such as protein depletion and viscosity reduction, with EV isolation, streamlining workflows for challenging fluids like synovial or bronchoalveolar lavage fluid [102].Tangential flow for analyte capture (TFAC) has emerged as a cutting-edge technique to purify extracellular vesicles from complex biological fluids, such as blood plasma and urine. TFAC employs ultrathin nanomembranes to selectively capture and concentrate EVs, which enhances the purity and yield compared to traditional methods like ultracentrifugation or size-exclusion chromatography. One of the main advantages of TFAC is its ability to process large volumes of biological samples while maintaining a high degree of selectivity for EVs. The tangential flow design reduces membrane fouling, allowing continuous flow and preventing sample loss. This leads to more efficient isolation of EVs, making it particularly useful for applications that require high-quality vesicles, such as biomarker discovery, drug delivery studies, and disease diagnostics. Its efficiency in preserving the integrity and functionality of EVs highlights its potential as a preferred method for EV purification in future studies [103].Acoustic waves isolation based on size and density is a pioneering, label-free method that leverages the principles of acoustofluidics—a fusion of acoustics and microfluidics. Acoustic-based EV isolation employs standing or traveling acoustic waves within a microfluidic channel to exert forces on particles suspended in the fluid. These acoustic forces guide EVs of specific size and density to defined locations, allowing size and density-based separation from other particles, such as cells, debris, and proteins. It is suitable for diverse biological fluids, including plasma, serum, urine, and cerebrospinal fluid. Acoustic isolation has been combined with molecular analysis platforms to streamline workflows in EV research and diagnostics [104].

**Figure 3 ijms-26-00189-f003:**
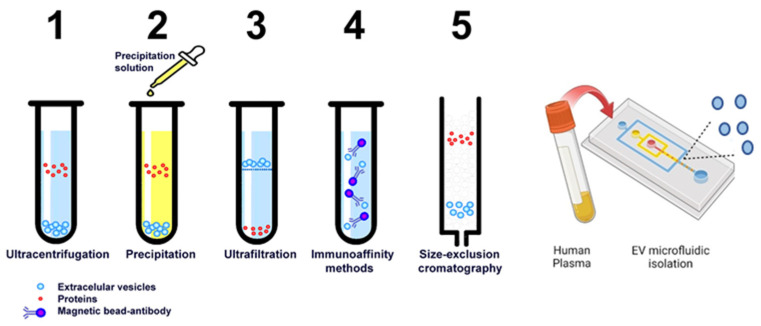
Common EV isolation methods.

**Table 3 ijms-26-00189-t003:** Extracellular vesicle isolation methods, detailing each method’s principle, advantages, limitations, and processing time.

Isolation Method	Principle	Advantages	Limitations	Processing Time
Differential Ultracentrifugation (UC)	Separates particles by density and size using high-speed centrifugal force.	High yield; widely used; inexpensive for labs with equipment.	Time-consuming; potential co-isolation of contaminants like proteins or lipoproteins.	~4–6 h
Density Gradient Centrifugation	Separates EVs based on density using a gradient medium (e.g., sucrose).	High purity; better separation from non-EV particles.	Requires skill to prepare gradients; low throughput.	~6–8 h
Size-Exclusion Chromatography (SEC)	Separates EVs from smaller molecules based on size via porous matrix.	Gentle on EVs; maintains functionality and integrity.	Moderate purity; potential loss of smaller EVs.	~1–2 h
Ultrafiltration	Filters EVs through membranes with specific pore sizes.	Fast and scalable; requires minimal equipment.	Potential clogging of filters; shear stress may damage EVs.	~30–60 min
Immunoaffinity Capture	Uses antibodies bound to solid supports to capture EVs expressing specific markers.	High specificity for target EV populations.	Expensive; limited to EVs with known surface markers.	~2–4 h
Precipitation-Based Methods	Uses polymers (e.g., polyethylene glycol) to precipitate EVs from solution.	Simple and fast; requires no specialized equipment.	Low purity; co-isolation of protein aggregates and polymers.	~1–2 h
Microfluidics-Based Isolation	Separates EVs using microfluidic devices based on size, charge, or affinity.	High precision; requires small sample volumes.	Expensive; limited throughput; requires specialized equipment.	~2–4 h
Affinity Chromatography	Isolates EVs based on binding interactions (e.g., lectins or ligands specific to EVs).	High specificity for certain EV populations.	Limited to EVs with known binding partners; relatively low yield.	~2–4 h
Tangential Flow Filtration (TFF)	Separates EVs by directing fluid tangentially across a filter surface.	Scalable; suitable for large volumes.	Requires optimization to minimize EV loss and contamination.	~2–3 h
Acoustic Wave Separation	Uses acoustic waves to isolate EVs based on size and density.	Label-free; minimal damage to EVs.	Requires specialized equipment; limited availability.	~1–2 h

Several commercially available products exist for extracellular vesicle isolation. ExoMir uses ultrafiltration with membranes of 200 nm and 20 nm pore sizes to trap particles for analysis. Precipitation-based kits like EXO-Prep and ExoQuick use reagents to isolate exosomes through low-speed centrifugation. Exo-spin Isolation Kit combines precipitation and size-exclusion chromatography techniques. As an example, Lobb et al. have shown that for the isolation of particles smaller than 100 nm, ExoQuick and Exo-spin products provided higher yield than qEV columns and OptiPrep DG methods [105]. Using human plasma, they also showed that although the qEV column had the lowest exosome recovery rate, it isolates exosomes with the highest puritySize-exclusion chromatography products, such as qEV and EVSecond, offer purity but vary in yield. Immunoaffinity-based methods, like Exosome-Human EpCAM and MagCapture, utilize antibodies or magnetic beads for cancer diagnostics. Membrane affinity kits, including ExoEasy, isolate EVs efficiently via spin columns. The ExoEasy Maxi Kit is a membrane affinity spin column method for exosome purification. Here, exosomes can be isolated from various samples, including plasma, serum, and cell culture supernatant, and the process takes only 25 min. Lastly, microfluidic devices like ExoChip immobilize antibodies to capture exosomes, leveraging their specific protein markers like CD63 [106].

The choice of EV isolation method depends on the intended application, balancing purity, yield, and scalability. Methods like ultracentrifugation and SEC are time-consuming and unsuitable for large-scale production. Polymer precipitation is more scalable but compromises purity and quality. Variability in experimental conditions (e.g., centrifugation speed, reagent quality) can lead to inconsistent results. The lack of universal standards for EV isolation and characterization complicates comparisons across studies. Non-vesicular contaminants (proteins, lipids, nucleic acids) are often co-isolated, especially in ultracentrifugation and precipitation methods. Reagents (e.g., PEG or antibodies) can introduce additional impurities, affecting downstream applications. In short, combining techniques (e.g., ultracentrifugation followed by SEC) often achieves better results. Addressing challenges like reproducibility and contamination will require standardized protocols and improved technologies to support both research and clinical applications.

### 5.3. Quality Control and Characterization

Standardized techniques such as nanoparticle tracking analysis (NTA), dynamic light scattering (DLS), tunable resistive pulse sensing (TRPS), and electron microscopy are used to measure the concentration and size distribution of isolated EVs. Flow cytometry (FC) or immunoblotting can be used to confirm the specificity and purity of isolated EVs [107,108]. EV morphology is currently best assessed for smaller EVs using high-resolution imaging techniques such as transmission electron microscopy (TEM) or scanning electron microscopy (SEM) cryo-EM and scanning-probe microscopy (SPM), including atomic force microscopy (AFM) [109,110]. AFM is essential for obtaining high-resolution morphological data, mechanical property measurements, and surface composition insights. While it is limited in throughput, it provides a unique depth of information that complements other EV characterization techniques [111].

FC is a key technology for the measurement of individual particles, but its application to the analysis of EVs has presented many challenges and has produced a number of controversial results, in part due to limitations of instrument detection, lack of robust methods, and ambiguities in how data should be interpreted.

### 5.4. Protein and Nucleic Acid Content

Standardized assays are used to quantify EV-associated proteins and nucleic acids, such as Western blotting, enzyme-linked immunosorbent assays (ELISA), or quantitative PCR (qPCR). Raman spectroscopy generates a unique “fingerprint” spectrum for the molecular content of EVs, revealing distinct molecular signatures associated with different EV subtypes or sources, allowing real-time tracking of EVs in physiological environments, which is valuable for understanding their biodistribution and interaction with target tissues [112].

Table 4 provides a quick reference for selecting EV analysis methods based on the specific information needed about the vesicles, sample limitations, and equipment available. Each method has unique strengths, allowing for a multifaceted approach to EV characterization when combined.

Mass spectrometry-based proteomic analysis and size analysis using nanoparticle tracking analysis (NTA) have demonstrated that size and composition heterogeneity extend to EVs derived from different cell types [113,114].

Single EV flow cytometry methods are gaining popularity due to their ability to characterize molecular cargo on individual vesicles. While most conventional flow cytometers are not configured to detect particles smaller than 500 nm in diameter, the instrumentation can be adapted to enable the detection of small particles [115]. The Amnis CellStream™ platform uses a time-delay integration charged-coupled device (TDI-CCD). Photon detection across spatially separated pixels allows for the capture of multiparametric image features, providing new dimensions of data [116]. They utilize increased detection sensitivity with pixel-based imaging features to resolve EV heterogeneity.

Recently, EVs have also been used as a platform to visualize and study enriched membrane proteins by cryoelectron transmission microscopy [117]. High-throughput technologies such as next-generation sequencing and mass spectrometry (proteomics, lipidomics, and transcriptomics), along with cryoelectron microscopy, contribute greatly to the evaluation of the molecular composition and structure of EVs [118].

Standardization is critical in EV research to ensure reproducibility, reliability, and comparability of results across studies. Both MIFlowCyt and MISEV frameworks play integral roles in the standardization of EV research [43,119]. MIFlowCyt focuses on standardizing data reporting and methodological transparency in flow cytometry, a commonly used technique for analyzing EV populations. This framework outlines the flow cytometer model and configuration, including laser lines, detectors, and threshold settings. Report acquisition settings, such as flow rate and total events, are recorded. It also provides gating strategies for identifying EV populations, ensuring clarity in distinguishing EVs from noise or contaminants. Including raw and processed data files with annotations enables re-analysis by other researchers. By adhering to MIFlowCyt, researchers can ensure that flow cytometry data on EVs is interpretable and reproducible, facilitating cross-laboratory comparisons [119].

The MISEV guidelines, established by the International Society for Extracellular Vesicles (ISEV), provide a broader framework for EV research. Updated periodically (most recently in 2023), MISEV emphasizes the following key areas: Define EV subtypes based on biophysical properties (size and density), biochemical markers, or cellular origin. Use multiple complementary techniques (e.g., differential centrifugation, density gradients) to isolate EVs [43]. Document the methods employed, including their limitations and potential co-isolation of non-EV components. Perform functional validation assays to confirm EV activity in biological contexts. Provide detailed protocols and datasets, emphasizing transparency and reproducibility. Share metadata, including experimental conditions, sample sources, and EV quantification methods. MISEV ensures that EV research adheres to high scientific standards, reducing variability across studies. Together, these frameworks enable researchers to produce reproducible, high-quality data, advancing the field of EV research and its therapeutic applications.

Lamparski et al. [120] have produced exosomes for a Phase I/II clinical trial by appropriately combining filtration methods, ultracentrifugation (UC), and sucrose gradient separation. They also developed quality control assays for the quantification and phenotypic characterization of exosomes. The ExoTEST™ kit, utilizing double-sandwich ELISA, was developed for the quantitative and qualitative analysis of exosomes.

In the future, an optimal technique for the large-scale production of clinical-grade EVs needs to be developed. EVs function as natural carriers of biomacromolecules, making them attractive candidates for the therapeutic delivery of various synthetic and biological molecules. EVs have been shown to offer advantages as delivery systems due to their nanoscale size, low immunogenicity, lack of cytotoxicity, and long-term safety.

## 6. EVS in Disease Diagnosis, Drug Delivery and Therapy Regenerative Medicine

Exosomes and microvesicles have garnered increasing interest in the field of biomedical research, particularly in diagnostics, gene therapy, drug delivery, and regenerative medicine [121]. This interest is primarily due to their ability to reflect the physiological or pathological state of the cells from which they originate. EVs offer a promising non-invasive tool for the early detection and diagnosis of various diseases. Their ability to carry disease-specific molecular cargo from damaged or stressed cells makes them highly valuable for real-time monitoring of disease progression, treatment response, and patient prognosis [122]. Although EVs are recognized as important communicators, their role in disease progression remains underexplored. Studies using human samples, tissue cultures, and animal models can help to reveal how EV-mediated communication affects the progression of various conditions.

### 6.1. Disease Diagnosis

EVs contain specific biomarkers that reflect the physiological or pathological state of the cells of origin. This makes them useful as diagnostic and prognostic tools for various diseases, including cancer, neurodegenerative disease, cardiovascular diseases, and immune and metabolic diseases (Table 5). Additionally, the detection of exosomes in body fluids such as urine, blood, and saliva could facilitate less invasive and more accessible diagnostics for early detection and monitoring of disease progression [122,123].

#### 6.1.1. Cancer

In cancer, EVs play a crucial role in promoting tumor progression by reprogramming immune cells, which aids in immune evasion and enhances cancer cell migration and proliferation. EVs derived from cancer cells have been associated with resistance to chemotherapy, as they help surviving cancer cells develop traits that increase their resistance to treatments. They also facilitate metastasis by helping circulating tumor cells (CTCs) adhere to distant tissues [124,125].

Research has identified several molecular mechanisms by which specific proteins within EVs contribute to cancer development, metastasis, and angiogenesis. Ral GTPases have been found to regulate metastasis by controlling the biogenesis of EVs and their organo-tropism—the ability of EVs to home in on specific tissues or organs. Ral GTPases influence the formation and release of EVs from cancer cells, directing these vesicles to distant metastatic sites, where they prepare the microenvironment for tumor colonization [126].

The tetraspanin Tspan8, commonly upregulated in tumor cells, plays a crucial role in altering the molecular content of tumor-derived EVs. Specifically, the expression of Tspan8 promotes the recruitment of nuclear proteins into EVs, which are then delivered to recipient cells, influencing their behavior [127]. This transfer of nuclear proteins can lead to modifications in gene expression and promote oncogenic processes in distant tissues. In addition to these mechanisms, the αvβ6 integrin, found in small EVs secreted by cancer cells, has been identified as a promoter of angiogenesis. These EVs stimulate pro-angiogenic signaling pathways in the tumor microenvironment, facilitating the formation of new blood vessels necessary for tumor growth and survival and promoting tumor expansion [128].

In the context of squamous cell carcinoma, Desmoglein 2 (Dsg2) has been implicated in the promotion of tumor growth via an EV-mediated mechanism. Dsg2 influences the EV cargo, notably altering the levels of IL-8 and miR-146a. This IL-8/miR-146a-dependent mechanism triggers pro-inflammatory and pro-tumorigenic responses, contributing to the establishment of a favorable microenvironment for tumor development and progression [129].

EVs play a pivotal role in cancer progression by transferring oncogenic molecules between tumor cells and other cells in the tumor microenvironment, which can contribute to tumor growth and metastasis. Two important aspects of EV-mediated cancer communication involve the activation of the Wnt signaling pathway and the modulation of immune responses through changes in cytokine levels. EVs containing oncogenic mutant β-catenin can transfer this mutated protein to recipient cells, where it activates the Wnt signaling pathway. Wnt signaling is crucial for cell proliferation and differentiation, and its dysregulation is commonly associated with tumorigenesis [130]. The presence of mutant β-catenin in EVs allows tumor cells to communicate with neighboring cells, promoting a pro-tumor environment. In recipient cells, these EVs initiate Wnt signaling by stabilizing β-catenin and enhancing its nuclear translocation, leading to the transcription of genes involved in cell proliferation and survival [67].

In addition to their role in signaling pathways, EVs from cancer cells also modulate the immune system. Cancer cell-derived EVs containing the alphaV beta6 integrin can significantly alter the behavior of peripheral blood mononuclear cells (PBMCs) by regulating the levels of CD163, IL-6, and IL-10. CD163 is a marker for macrophage activation, while IL-6 and IL-10 are key cytokines involved in inflammation and immune suppression, respectively [131]. By increasing the levels of these molecules in PBMCs, cancer-derived EVs promote an immunosuppressive environment that favors tumor growth and metastasis. This immunomodulatory effect, mediated by EVs, helps cancer cells evade immune detection and enhances the inflammatory responses that support tumor progression [132].

The contents of EVs in cancer cells are often altered, creating unique molecular signatures that help differentiate healthy tissue from cancerous tissue. In particular, miRNAs within EVs are important because they remain stable, and their expression patterns can be linked to specific cancer types and stages [133].

Circulating EVs are typically found in elevated numbers in cancer patients. Their molecular profiles offer real-time information about tumor dynamics without requiring invasive tissue biopsies. Research has shown that EVs released by early-stage tumors can carry specific biomarkers, enabling the detection of cancer even before clinical symptoms emerge. Because EVs carry cargo from various tumor regions, they offer a more comprehensive view of tumor biology than traditional biopsies [134,135].

#### 6.1.2. Neurodegenerative Diseases

EVs play a pivotal role in neurodegenerative diseases by facilitating intercellular communication and contributing to disease progression and diagnosis. EVs are involved in transporting toxic proteins, such as tau, amyloid-beta, and alpha-synuclein—proteins that are closely associated with neurodegenerative disorders like Alzheimer’s and Parkinson’s disease [136,137]. These misfolded proteins, carried by EVs, help in diagnosing and monitoring the progression of these conditions as they can spread pathology between neurons and glial cells, contributing to the deterioration across different brain regions. Additionally, EVs create a pro-inflammatory environment that exacerbates neurodegeneration and neuronal loss.

EVs also carry disease-specific molecular signatures, including altered microRNAs (miRNAs) that are linked to Alzheimer’s and Parkinson’s, making them valuable biomarkers for early diagnosis when timely intervention is critical. In conditions like Huntington’s disease, EVs can carry mutated huntingtin protein, detectable in biofluids such as cerebrospinal fluid (CSF) and blood, aiding in diagnosis [136,137].

Glial cells, including microglia and astrocytes, secrete EVs that carry inflammatory mediators, such as cytokines, which drive chronic neuroinflammation in neurodegenerative diseases. Dysfunctional EV signaling contributes to oxidative stress and additional neuronal damage, as activated microglia release EVs containing pro-inflammatory molecules and damaged proteins, further promoting neuroinflammation and degeneration [138].

#### 6.1.3. Cardiovascular Diseases

Extracellular vesicles are increasingly recognized as vital biomarkers for the early detection of cardiovascular diseases, including atherosclerosis, heart failure, myocardial infarction, and hypertension [139].

Elevated numbers of circulating EVs, particularly those derived from endothelial cells, platelets, and leukocytes, have been associated with increased cardiovascular risk and guiding therapeutic interventions [140]. EVs carry markers of inflammation, coagulation, and oxidative stress, which are closely linked to the pathophysiology of cardiovascular diseases. EVs, in this context, contribute to a pro-thrombotic environment by carrying coagulation factors, tissue factors, and phosphatidylserine, all of which enhance blood clot formation. The correlation between high levels of pro-thrombotic EVs and increased risk of thrombosis suggests that EVs play a crucial role in the development of thrombotic events such as myocardial infarction and stroke. Monitoring the levels and content of circulating EVs could, therefore, be a predictive marker for identifying individuals at higher risk of cardiovascular events [141].

After myocardial infarction, cardiomyocytes release EVs that promote inflammation, fibrosis, and tissue remodeling. Some EVs, such as those containing miR-126, assist in heart repair by stimulating angiogenesis and regenerating heart tissue. EVs are also involved in regulating cardiac hypertrophy, a condition where the heart muscle thickens, potentially leading to heart failure. In heart transplant patients, exosomes can help monitor for organ rejection or infection [142].

In hypertension, EVs from immune cells, endothelial cells, and vascular smooth muscle cells exacerbate vascular inflammation and oxidative stress. They carry reactive oxygen species (ROS) and pro-oxidative enzymes like NADPH oxidase, contributing to endothelial dysfunction and blood pressure elevation. EVs involved in the renin-angiotensin system also play a role in vasoconstriction, further increasing blood pressure. Additionally, EVs facilitate thrombosis by carrying pro-coagulant factors, including tissue factor (TF) and phosphatidylserine, released by platelets and endothelial cells [143].

Recent research has focused on identifying protein components within cardiac progenitor cell (CPC)-derived EVs that mediate cardiac repair. EVs secreted by CPCs have shown the ability to transfer bioactive molecules to damaged cardiac tissue, promoting repair and activating regenerative signaling pathways, reducing apoptosis, and enhancing the proliferation of cardiac cells [144].

#### 6.1.4. Immunological and Autoimmune Diseases

Extracellular vesicles play a crucial role in regulating the immune system by reflecting immune activation and contributing to autoimmune disease progression. These vesicles facilitate communication between key immune cells, including T cells, B cells, dendritic cells, and macrophages [145]. Depending on their origin and cargo, EVs can either enhance or suppress immune responses. For instance, dendritic cell-derived EVs can present antigens to T cells, boosting immune activation and maintaining immune surveillance. On the other hand, certain EVs carry immunosuppressive molecules that reduce immune responses and promote tolerance.

EVs derived from different sources, such as amniotic fluid stem cells (AFSCs) and CD4+ T regulatory cells (Tregs), play significant roles in modulating immune responses. These EVs are particularly relevant in autoimmune conditions, where immune dysregulation leads to the loss of immune tolerance and neuronal damage [145]. In experimental autoimmune encephalomyelitis, an animal model for multiple sclerosis, AFSC-derived EVs were found to reduce the severity of the disease by suppressing pro-inflammatory immune responses [146]. These vesicles carry a variety of immunomodulatory molecules that can inhibit the activation of pro-inflammatory T cells and macrophages, thereby reducing neuroinflammation. Studies suggest that AFSC-derived EVs can shift the immune response from a pro-inflammatory state towards an anti-inflammatory one, leading to a reduction in the demyelination and neuronal damage typically seen in multiple sclerosis [147].

EVs derived from human Tregs have been found to play a vital role in cell-to-cell communication and the maintenance of immune homeostasis. These vesicles contain specific microRNAs, such as miR-142-3p and miR-150-5p, which regulate immune pathways and can influence the behavior of target immune cells. In the context of multiple sclerosis, the loss of immune tolerance is a central issue, and studies have shown that a dysfunction in Treg-derived EVs may contribute to this loss. The miRNAs carried by Treg-derived EVs are involved in suppressing the activation of pro-inflammatory T cells and promoting the expansion of other regulatory T cells. When the function or release of these EVs is impaired, it may lead to the immune system attacking myelin, as seen in multiple sclerosis, thus contributing to disease progression [148]. Tregs produce IL35-coated EVs that contribute to immune tolerance. These vesicles suppress the activation of effector T cells and promote infectious tolerance, helping to regulate immune responses in inflammatory diseases. [149].

In chronic Chagas disease, EVs carry parasite-derived molecules that modulate host immune responses, contributing to chronic inflammation and disease persistence. The presence of these circulating EVs offers insights into disease mechanisms and suggests a potential role for EV-based diagnostics or therapies in Chagas disease [150].

Exosomes released during infections caused by pathogenic Gram-negative bacteria, such as *Escherichia coli*, play a critical role in modulating innate immune responses. These exosomes are known to carry bacterial components, such as LPS, which activate immune signaling pathways, including Toll-like receptors [151]. The human pathogenic fungus *Candida albicans* has been shown to induce the release of immunomodulatory vesicles from immune cells, specifically carrying transforming growth factor-beta 1. These vesicles promote immune tolerance and help the pathogen evade immune surveillance [152]. The interaction between vesicle cargo and immune cells can either enhance inflammatory responses or contribute to immune evasion, depending on the context of the infection.

#### 6.1.5. Metabolic Diseases

In metabolic diseases, EVs play a pivotal role by modulating key processes such as insulin sensitivity, lipid metabolism, and inflammation [153,154,155]. In obesity, adipocytes secrete EVs loaded with pro-inflammatory molecules that contribute to systemic inflammation and the development of insulin resistance. These EVs also impair the function of muscle and liver cells, disrupting glucose uptake and lipid metabolism [155].

In type 2 diabetes (T2D), EVs further exacerbate insulin resistance by impacting pancreatic beta-cell function. Elevated levels of circulating EVs are observed in T2D patients, particularly those originating from immune cells, adipocytes, and endothelial cells [154]. These vesicles often carry inflammatory cytokines like IL-6 and TNF-α, which interfere with insulin signaling pathways, contributing to insulin resistance.

EVs are also critical in the progression of non-alcoholic fatty liver disease (NAFLD). Under stress from fat accumulation, hepatocytes release EVs containing pro-inflammatory molecules, as well as miR-122 and miR-192, which promote liver inflammation and fibrosis. These vesicles activate immune cells, worsen liver damage, and contribute to disease progression by influencing lipid metabolism and fibrogenesis. Altered EV content in NAFLD can reflect liver dysfunction and act as biomarkers for early diagnosis and disease severity. EVs have also been found to activate hepatic stellate cells, promoting liver fibrosis through the delivery of pro-fibrotic signals [156].

One of the most promising applications of exosomes and microvesicles is their use as vehicles for drug and gene delivery. Due to their ability to protect biological content from degradation in the extracellular environment and efficiently deliver it to specific cells, these vesicles are ideal for gene therapy [22]. EVs can encapsulate and protect genetic material for their transport, facilitating their delivery to specific cells. This is particularly useful in gene therapy, where the goal is to correct or modify gene expression to treat diseases [157,158]. Exosomes containing siRNA can be designed to silence specific genes associated with diseases, such as cancer, where silencing oncogenes could inhibit tumor growth.

These EVs can be used for targeted drug delivery, increasing treatment efficacy and minimizing side effects. Their ability to fuse with the plasma membrane of recipient cells allows for the direct release of their contents into the cytoplasm. Exosomes protect drugs from premature degradation in the bloodstream, increasing bioavailability and prolonging the drug’s half-life. This is crucial for drugs with rapid metabolic degradation [159].

Effective administration strategies are crucial to maximizing their therapeutic efficacy, ensuring target specificity, and overcoming challenges related to their stability and biodistribution. Systemic delivery involves introducing EVs into the bloodstream via intravenous (IV) or intraperitoneal injection [160]. This approach is commonly used for therapeutic interventions that target widespread or inaccessible areas, such as metastatic cancers or neurodegenerative diseases. While systemic delivery enables EVs to reach various tissues, it faces challenges such as rapid clearance by the mononuclear phagocyte system (MPS), off-target effects, and variable biodistribution. Strategies to enhance systemic delivery include engineering EVs with surface modifications, such as PEGylation, to increase circulation time and functionalizing them with ligands or antibodies to enhance tissue-specific targeting [161].

Localized delivery involves administering EVs directly to the site of injury or disease, such as through intra-articular injections for joint disorders, intratumoral injections for cancer, or topical application for wound healing. This method minimizes systemic side effects and enhances therapeutic concentration at the target site. For example, EVs derived from mesenchymal stem cells have shown promise in promoting cartilage repair when delivered locally. Local administration is advantageous for precision and efficiency but may be limited by the accessibility of the target site and the potential for uneven distribution within the tissue [162].

Though less common, oral administration of EVs is being explored, particularly for EVs derived from plant or milk sources. These EVs exhibit high stability in the gastrointestinal environment and can deliver bioactive components to the gut or systemic circulation [163]. Challenges include potential degradation by digestive enzymes and variable absorption rates. Nasal delivery of EVs has emerged as a novel route to target the central nervous system. This strategy exploits the direct connection between the nasal cavity and the brain via the olfactory and trigeminal nerves. EVs delivered intranasally have demonstrated the ability to bypass the blood-brain barrier and deliver therapeutic cargo for conditions such as Alzheimer’s disease or Parkinson’s disease [164]. This non-invasive approach is gaining traction for its efficiency and ease of application.

To optimize the therapeutic potential of EVs, advanced engineering techniques are employed. Surface modifications using peptides, antibodies, or aptamers can enhance the specificity of EVs to particular tissues or cell types. Additionally, pre-loading EVs with therapeutic agents, such as small RNAs or drugs, can be achieved using methods like electroporation, sonication, or incubation. These strategies enable the customization of EVs for diverse therapeutic applications. The choice of administration route depends on the therapeutic target, desired biodistribution, and procedural risks in a disease context, and determining the optimal dose and dosing strategy is complex. Continued advancements in EV engineering and delivery technologies hold promise for unlocking their full potential in clinical applications [165].

In cancer treatment, exosomes can be loaded with chemotherapeutic agents and specifically targeted to tumor cells, reducing systemic toxicity and improving drug concentration at the tumor site. Exosomes loaded with neuroprotective or anti-inflammatory drugs can cross the blood-brain barrier, making them promising vehicles for treating diseases like Alzheimer’s and Parkinson’s [136,137].

EVs offer several key advantages in fighting viral infections, such as delivering antiviral agents, modulating immune responses, and blocking viral replication [166]. EVs naturally travel to specific cells, allowing targeted delivery to infected tissues reducing off-target effects. EVs can cross the blood-brain barrier, making them useful for treating viral infections in the central nervous system (e.g., HIV, herpes) [167]. Their natural biocompatibility reduces the risk of toxicity compared to synthetic drug carriers. Some viruses can bind to cellular receptors via proteins expressed on EV surfaces. EVs can act as “decoys,” preventing viruses from binding to their target host cells, thereby reducing viral entry and replication. EVs can bind to viral particles, inhibiting their ability to attach to immune cells and reducing infection rates [168].

Some viral infections, such as HIV, can use EVs to deliver immunosuppressive molecules, aiding viral persistence. EVs can be designed to carry neutralizing antibodies or receptor blockers to prevent viral entry into cells, disrupting the viral life cycle [168].

In anti-inflammatory therapy, EVs have a fundamental role due to their role in modulating immune responses and tissue repair. EVs can be engineered to carry anti-inflammatory molecules and therapeutic proteins. So, specific miRNAs such as miR-146a and miR-21 can suppress pro-inflammatory pathways, reducing inflammation in diseases such as rheumatoid arthritis and inflammatory bowel disease [169]. EVs loaded with anti-inflammatory cytokines like interleukin-10 (IL-10) can reduce inflammation by modulating immune responses, leading to tissue healing and lessening of inflammation [59].

EVs derived from MSCs and other cell types have demonstrated the ability to modulate immune activity and reduce inflammation in autoimmune diseases. Beyond modulating inflammation, EVs also play a role in tissue repair by promoting angiogenesis and reducing fibrosis [170].

Immunomodulatory MSC-derived small EVs have demonstrated potential as cell-free therapies for both acute and chronic pulmonary vascular diseases. By modulating immune responses, these vesicles can reduce pulmonary hypertension, control vascular remodeling, and improve lung tissue repair in conditions like chronic obstructive pulmonary disease (COPD) and pulmonary fibrosis [171]. Pro-inflammatory priming of umbilical cord MSCs alters the protein cargo of their extracellular vesicles. These changes in protein content enhance their anti-inflammatory and regenerative capacities, improving their efficacy in treating inflammatory diseases. The ability to modulate EV content through priming techniques opens new possibilities for tailored EV-based therapies [172].

Monocytes have been shown to traffic extracellular vesicles to damaged muscle, where they adopt a novel immunophenotype that supports muscle regeneration. MSC-EVs help reprogram monocytes, enhancing their reparative functions and promoting tissue recovery after injury. This mechanism holds promise for therapeutic approaches in muscle injuries and degenerative diseases [173]. MSC-derived extracellular vesicles promote cartilage regeneration by regulating autophagy in chondrocytes. By controlling the autophagic process, MSC-EVs enhance cartilage repair, offering potential treatments for degenerative joint diseases like osteoarthritis [174].

In cardiology, exosomes derived from MSCs have shown the potential to promote cardiac tissue regeneration after myocardial infarction. These exosomes can reduce cell apoptosis, stimulate angiogenesis (formation of new blood vessels), and improve cardiac function [175].

Evaluating the immunogenicity and long-term effects of exosome-based therapies is also essential for their clinical development. This highlights the significance of MSC-EVs in tissue engineering and regenerative medicine.

## 7. Challenges and Future Directions in Extracellular Vesicles Research

Despite the promise of EVs in clinical applications, several challenges remain in their bioengineering, purification, storage, and regulatory compliance. Isolating pure EVs from complex biological fluids remains a significant challenge. Standardization of purification protocols and the development of more efficient, scalable methods are critical for advancing EV-based diagnostics and therapeutics [176]. EVs are inherently unstable and sensitive to environmental conditions such as temperature, pH, and proteolytic degradation. Storing EVs without compromising their integrity and functionality is an ongoing issue, particularly for clinical use. Researchers are exploring cryopreservation and lyophilization techniques to maintain EV stability during long-term storage, but more work is needed to optimize these methods for large-scale production and storage. The clinical use of EVs, particularly for therapeutic purposes, faces stringent regulatory hurdles. Since EVs are biological entities, they are subject to the same regulatory scrutiny as cell-based therapies. The Food and Drug Administration and other regulatory bodies are in the process of developing guidelines for EV-based therapeutics, but standardized testing and manufacturing protocols remain limited. There is also a lack of consensus on how to classify and manufacture EV-based therapies, which can delay their approval for clinical use [176,177].

EVs display inherently clinically desirable characteristics for therapeutic as the ability to contain diverse biomolecular cargos, the ability of said cargo to elicit potent cellular responses, to cross biological barriers, availability, bioengineerability, and scalability [178]. In contrast to their tremendous potential, only a few EV-based therapies and drug delivery have been approved for clinical use, which is largely attributed to limited therapeutic loading technologies and efficiency.

The natural heterogeneity of EVs poses significant challenges for their use in both diagnostics and therapeutics. Disease markers are sometimes restricted to specific EV subpopulations, which can reduce diagnostic accuracy if these subpopulations are not well identified or isolated. Similarly, in therapeutic contexts, certain EV subgroups may deliver the desired therapeutic effects, while others might be ineffective or even adverse, complicating the assessment of efficacy and safety in clinical trials [179]. Further complicating this issue is the difficulty in distinguishing between various EV subtypes, which share overlapping physical characteristics. This complexity is often overlooked, as many studies refer to “exosomes” broadly without recognizing the distinct functionalities and compositions across the full range of EV subtypes involved [180].

Significant challenges remain in the field of EV research that must be addressed to fully realize their clinical and therapeutic utility [181] (Figure 4).

Standardization is essential for translating EV-based therapies into clinical practice. Variability in EV isolation, characterization, and quantification poses a significant challenge, as inconsistent quality can affect both safety and efficacy. Emerging methods, such as microfluidics and tangential flow filtration, are promising but require further optimization and validation. Variations in protocols between laboratories lead to inconsistent results, complicating the interpretation of EV research. To address these issues, international organizations are developing guidelines for EVs, ensuring that the methods used are reproducible and yield consistent EV populations [43]. However, more widespread adoption of these guidelines and refinement of best practices are required to facilitate comparison across studies and accelerate translational research. Issues such as large-scale production, ensuring safety, and avoiding immunogenicity must be addressed. The development of good manufacturing practice-compliant methods for producing clinical-grade EVs is critical for moving EV-based therapies from the laboratory to the practice clinic.

The functional roles of EVs in various physiological and pathological processes are still not fully understood. Although EVs are known to participate in immune modulation, cancer progression, tissue repair, and cell signaling, the precise molecular mechanisms by which they exert their effects remain unclear. Functional assays for EVs aim to elucidate their biological roles and therapeutic potential by analyzing how EVs interact with recipient cells. The implementation of these assays typically follows a structured workflow, from EV isolation to analysis of their effects on target cells or organisms. Before performing functional assays with EVs, the first step is isolating high-quality EVs from sources like cell culture media, body fluids, or tissues. After isolation, the EVs are characterized for purity and concentration [182]. For tracking and biodistribution, EVs are labeled with fluorescent dyes or genetic markers. Functional assays evaluate how EVs are internalized by recipient cells, influencing processes like cell survival, proliferation, apoptosis, and migration, which are crucial in tissue repair and cancer metastasis [183]. Gene expression changes induced by EV cargo can be tracked using reporter assays. Additionally, EVs can be loaded with therapeutic molecules for in vitro or in vivo delivery, and drug release is monitored in target cells. More advanced models and in vivo studies are needed to unravel these mechanisms and assess the long-term impact of EVs on disease progression and treatment. Recent advancements have shown that engineered EVs can deliver specific RNA or protein-based drugs, such as small interfering RNA (siRNA) or messenger RNA (mRNA), directly to target cells. These EVs can be engineered to improve loading efficiency, target specificity, and therapeutic activity [56].

Extracellular vesicles are promising vehicles for therapeutic applications due to their biocompatibility, stability, and inherent ability to transfer cargo to specific cells or tissues. Efficient techniques for loading cargo, such as therapeutic RNAs, proteins, or small molecules, into EVs have been developed. These techniques can be broadly categorized into passive and active loading strategies, each with specific advantages and limitations [184,185].

Passive loading relies on the natural incorporation of cargo into EVs during their biogenesis or isolation. Donor cells are genetically modified or treated with cargo of interest, which is naturally packaged into EVs during biogenesis. They have high specificity for endogenous EV pathways and are time-intensive and dependent on cell viability. Cargo molecules are incubated with EVs, allowing diffusion or spontaneous association with the vesicles. It presents low loading efficiency and challenges in delivering larger cargoes. This method is simple and non-disruptive to EV structure.Active loading employs external forces or treatments to enhance cargo incorporation, often disrupting the EV membrane temporarily to facilitate loading. By electroporation, high-voltage electric pulses create temporary pores in EV membranes, enabling the entry of large or charged cargo molecules, such as siRNAs or plasmids. It is suitable for a wide range of molecules, including nucleic acids, and may cause EV aggregation or cargo degradation. Sonication ultrasound waves disrupt the EV membrane, allowing cargo to diffuse inside. Membranes reseal after the process. It presents high loading efficiency for hydrophilic and hydrophobic molecules and potential alteration of EV structure and function. EVs and cargo are forced through nanoporous membranes, causing membrane fusion and integration of the cargo by extrusion. It is effective for larger molecules and may damage EV integrity. Repeated freezing and thawing cycles disrupt EV membranes, allowing cargo incorporation. It presents a risk of vesicle damage and aggregation, and this is simple and cost-effective. Surfactants or membrane-disrupting agents (e.g., saponin) temporarily permeabilize EV membranes for cargo incorporation. Enhances loading for certain drugs and has the potential for irreversible damage to EVs [56,184].

EV cargo loading techniques face several challenges, so variability in EV size, composition, and cargo capacity affects loading consistency. Contamination with residual cargo molecules outside EVs can complicate downstream applications. Many methods, especially active techniques, are difficult to scale for clinical applications. Some methods risk degrading sensitive biomolecules, such as mRNAs or proteins [185].

Recent research focuses on improving the efficiency and specificity of EV cargo loading. Cargo loading is a critical step in the therapeutic application of EVs, and continuous improvements in methodologies promise to enhance their clinical and research utility. Modified donor cells or synthetic EV mimics are tailored for enhanced cargo incorporation. Combining multiple loading techniques, such as sonication with electroporation, to optimize efficiency. Surface engineering of EVs to improve targeting and stability post-loading.

Additionally, do not have a reliable means to quantify the amount of cargo loaded into an EV. It was long thought that exosomes derived from the same cells had similar compositions of proteins, nucleic acids, and lipids, but recent reports have shown that exosomes from the same parent cells can have different molecular compositions. Not all EVs isolated from a given modified cell line have the same targeting motif [186,187]. Therefore, a better understanding of the heterogeneity and molecular composition of EVs could allow us to identify subpopulations more suitable for certain EV-based therapies (i.e., identifying subpopulations that can exert particular effects without undesirable side effects).

Researchers have developed methods to modify the surface of exosomes with fusogenic proteins that facilitate their binding and fusion with specific cellular membranes. Fusogenic exosomes are engineered to possess specific proteins or lipids on their surface that promote membrane fusion [188]. These modifications often involve incorporating fusogenic viral proteins or using naturally fusogenic lipids. The fusion process typically begins with the recognition and binding of the exosome to the target cell membrane, followed by the fusion of the lipid bilayers, resulting in the direct delivery of exosomal content into the recipient cell’s cytoplasm [189]. Fusogenic exosomes loaded with soluble cargos, such as transcription factors or cytosolic proteins, can be used to alter or supplement the biological pathways of recipient cells, emerging as a promising frontier and opening new avenues for cellular therapy and genetic engineering.

Extracellular vesicles derived from food sources, such as plants and milk, are gaining recognition as key players in the development of next-generation functional foods. These nanoscale vesicles, naturally secreted by cells in plants and other dietary sources, act as efficient nanocarriers for a wide range of bioactive including polyphenols, flavonoids, and other secondary metabolites [190]. Studies on these vesicles have demonstrated that they can be extracted from various plant components, such as juices, seeds, and even dried plant materials [191], and are characterized based on properties such as density, size, cellular or environmental origins, and biochemical composition, highlighting their diverse nature and potential applications [190]. Unlike in mammals and yeasts, the formation of plant-derived EVs is thought to occur through the maturation of clathrin-coated tubular networks (TGN) within the Golgi stack matrix, a process functionally analogous to the role of early endosomes (EE) in mammalian cells. Research findings provide evidence suggesting that plant EVs originate from TGN/multivesicular bodies [192].

An interesting aspect of this review is based on the structural stability and uptake pathways of food-derived EVs (FDEVs) to target cells and their health benefits, including antioxidant, anti-inflammatory, anticarcinogenic effects, gut microbiome modulation, and intestinal barrier enhancement [193]. These EVs are shielded from gastrointestinal degradation and enzymatic breakdown, significantly enhancing their stability and bioavailability. Furthermore, possess inherent biocompatibility and low immunogenicity, making them ideal candidates for oral delivery systems in functional food applications. Their dual role as stabilizers and carriers of bioactive compounds positions them at the cutting edge of innovative strategies for improving human health through dietary interventions [190]. However, the biological functions of FDEVs are not fully understood, and standard isolation protocols are lacking.

Milk-derived EVs enhance intestinal barrier integrity and modulate gut microbiota composition, reducing gut inflammation. They carry immune-regulatory microRNAs (e.g., miR-148a) that influence immune response and tolerance mechanisms. Specific examples include plant-derived EVs (e.g., ginger and grapes) with anti-cancer and antioxidant properties and milk-derived EVs that support wound healing and tissue repair [194]. MiRNAs within the EVs enhance intestinal permeability, support epithelial cell modification, and modulate immune responses. The internalization of orange juice-EVs into intestinal epithelial cells is able to modulate the expression of important genes related to inflammatory pathways, such as *HMOX-1* and *ICAM1*, or to the restoration of intestinal permeability related such as claudins and occludin [195]. Orally administered fruit-derived EVs activate Wnt/-catenin signaling in intestinal stem cells, promoting cell proliferation and intestinal wall integrity. Endocytosis studies reveal that food-derived EVs are absorbed by intestinal cells and macrophages, influencing local gut properties and reaching distal organs via circulation to affect systemic health [196]. FDEVs reduce pro-inflammatory cytokines (IL-1β, IL-6) and increase anti-inflammatory IL-10 levels, demonstrating efficacy in mouse models of inflammatory bowel disease. These findings suggest promising therapeutic potential for conditions like Crohn’s disease and ulcerative colitis in humans [197,198].

Integrating Organ-on-a-Chip (OoC) technology with AI offers promising advancements for extracellular vesicle (EV) research, enhancing reproducibility and precision. AI can process the extensive datasets generated by OoCs, identifying complex patterns in EV interactions and dynamically adjusting experimental parameters [199]. This functionality not only improves data interpretation but also helps standardize results across labs, addressing challenges like experimental bias due to the intricacies of isolating and categorizing EVs consistently [200].

In diagnostics and therapeutics, AI-enhanced OoC systems enable scalable, rapid EV analysis, supporting more personalized medicine [201]. For instance, machine learning algorithms could predict the therapeutic potential of specific EV subtypes for targeted conditions or customize EV-based interventions using patient-specific data. Such approaches are particularly relevant for complex diseases—cancer, cardiovascular, and neurodegenerative disorders—where EVs could be engineered for precise delivery with minimized off-target effects [202]. AI can also assess EV transport across tissue barriers, interpret EV-induced gene expression changes, or pinpoint biomarkers for early disease detection, accounting for the physiological complexity modeled in OoCs.

OoCs further enable the study of organ-specific EV release and interactions. In liver models, for example, OoCs can simulate dynamic hepatic EV release under flow conditions that mimic liver function. In cancer research, tumor-on-chip platforms allow researchers to explore EV-mediated communication between tumor cells and their microenvironment [203,204]. These controlled setups facilitate studies on how EVs influence tumor progression, immune modulation, and drug resistance. By incorporating EVs into OoCs, scientists gain real-time insights into EV-based drug delivery, including pharmacokinetics, efficacy, and safety, which can inform more personalized treatment strategies and reduce adverse effects in clinical applications. Additionally, OoCs have potential in regenerative medicine, where EVs can support tissue repair and regeneration within organ-mimicking environments. This could facilitate therapeutic applications by leveraging EVs as multi-omic molecular reservoirs, well-suited for analysis through AI-driven approaches that are already advancing therapeutic development and the understanding of disease mechanisms.

The future of EV research, therefore, lies in an interdisciplinary approach that combines bioengineering, computer science, and molecular biology. Advancements in OoC and AI will likely accelerate the translation of EV research into clinical practice, promoting the way for highly personalized, efficient, and scalable EV-based diagnostics and therapy tools in precision medicine.

## 8. Conclusions

EVs reflect the physiological or pathological states of their cells of origin, making them valuable biomarkers for early disease detection and monitoring, carrying promise in non-invasive diagnostics. Integrative proteo-transcriptomic analyses represent a frontier in EV research, bridging molecular biology and clinical application to advance our understanding of EV-mediated processes and their translational potential in medicine. The molecular contents of EVs can reveal tumor characteristics and track treatment responses. In neurodegenerative and cardiovascular diseases, EVs provide a window into disease progression and patient health, aiding in timely and accurate diagnosis. The innovative use of EVs in crossing biological barriers, such as the blood-brain barrier, showcases their potential in treating conditions like Alzheimer’s and Parkinson’s diseases. EVs can reduce inflammation, enhance angiogenesis, and support cell survival, offering new therapeutic avenues in conditions like heart disease, nerve injuries, and musculoskeletal disorders. EVs naturally travel to specific cells, allowing targeted delivery to infected tissues reducing off-target effects.

Extracellular vesicles have significant potential in diagnostics and therapeutics due to their natural role in intercellular communication and targeted delivery capabilities. However, challenges in EV heterogeneity, therapeutic loading, and consistent quality currently limit their clinical application. Advancements in standardizing EV isolation, characterization, and quantification are critical to achieving reproducible and precise outcomes. Food-derived extracellular vesicles represent a novel link between nutrition and therapeutics, serving as natural nanocarriers of bioactive molecules and metabolites. Integrating EV research with organ-on-chip platforms and artificial intelligence could help overcome these barriers by enabling accurate modeling of EV interactions in physiological settings and optimizing personalized therapeutic approaches. With continued innovation in scalability, safety, and regulatory standards, EVs could become central to precision medicine and targeted therapies.

## Figures and Tables

**Figure 1 ijms-26-00189-f001:**
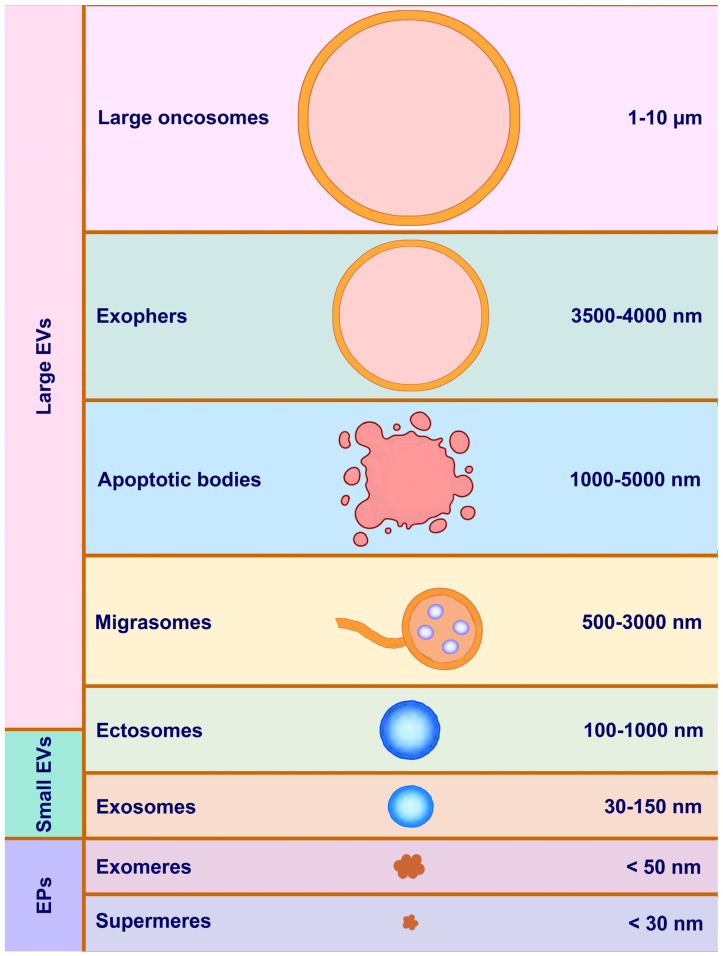
Subtypes of EVs include exosomes, ectosomes/microvesicles, migrasomes, apoptotic bodies, exophers, and large oncosomes. Cells also secrete extracellular particles (EPs), including exomeres and supermeres. Redrawn from www.microvesicles.org.

**Figure 2 ijms-26-00189-f002:**
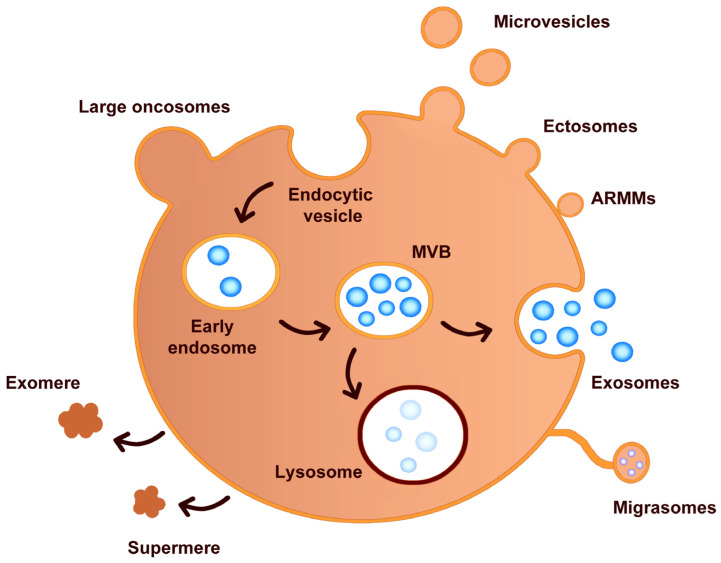
Different types of extracellular vesicles produced by the cell. These vesicles are categorized based on their mode of synthesis and size. Redrawn from Stahl et al. [29].

**Figure 4 ijms-26-00189-f004:**
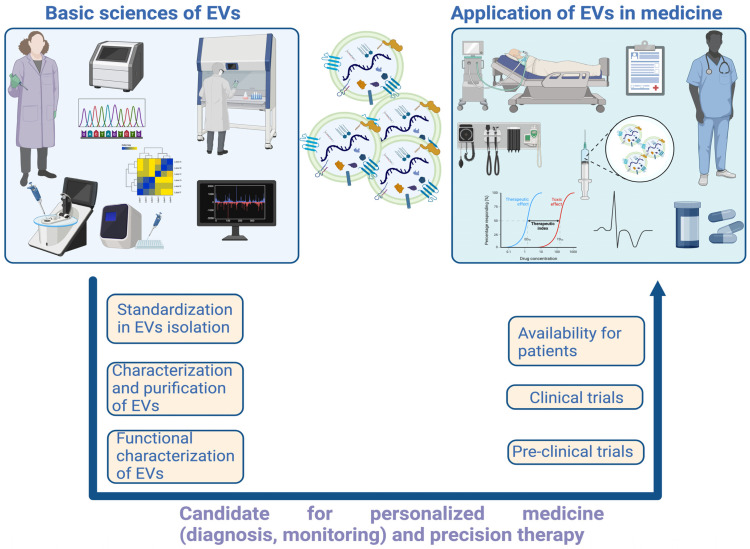
Challenges and future directions in extracellular vesicles research.

**Table 1 ijms-26-00189-t001:** Classification of vesicles based on size, biogenesis, and markers.

Extracellular Vesicles	Size	Biogenesis	Markers
Exosome	30–150 nm	Endosomal route	CD9, CD63, CD81, Alix, TSG101, HSP70, HSP90β, flotillins
Microvesicles	100–1000 nm	Budding of plasma membrane	Selectins, ARF6, CD40, cytoskeletal protein, heat shock proteins, integrins, flotillins
Apoptotic Bodies	1000–5000 nm	Separation of plasma membrane from the cytoskeleton	Histones, HSP60, GRP78, proteomic profile similar cell lysate

**Table 2 ijms-26-00189-t002:** EVs databases.

Database	Content	Web Site
Vesiclepedia	Provide information of molecular contents data (lipids, metabolites, nucleic acids and proteins, RNA miRNA origins, and associated biological functions	htpps://www.microvesicles.org (accessed on 22 December 2024)
ExoCarta	Focuses on exosomes, a specific type of extracellular vesicle. It catalogs proteins, RNA, and lipids components identified in exosomes. Datasets including experimental methods and conditions isolate and analysis	htpps://exocarta.org (accessed on 22 December 2024)
EVpedia	Integrative database for EV research, providing access to data on proteins, RNAs, lipids, and metabolites associated with EV. It offers tools for data analysis to explore data in the context of EV biology	https://evpedia.info/evpedia2_xe/ (accessed on 22 December 2024)
ExoRBase	Exosomal RNA, particularly non-coding RNA like miRNAs, lncRNAs, and circRNAs. Data on RNA sequences, expression profiles, and potential functions, making it useful for studying the regulatory roles of exosomal RNA.	htpps://exorbase.org (accessed on 22 December 2024)
miRandola	Data on circulating miRNAs that are found in body fluids, both free and within EVs. miRNA sequences, associated diseases, and experimental evidence for their presence in EVs.	https://ngdc.cncb.ac.cn/databasecommons/database/id/4321 (accessed on 22 December 2024)
EVAtlas	A comprehensive database for ncRNA expression in EVs with functional modules.	http://bioinfo.life.hust.edu.cn/EVAtlas (accessed on 22 December 2024)

**Table 4 ijms-26-00189-t004:** Highlights the trade-offs between sensitivity, throughput, and processing time for commonly used EV analysis methods.

Method	Principle	Sensitivity	Throughput	Processing Time
Nanoparticle Tracking Analysis (NTA)	Tracks Brownian motion of particles to estimate size and concentration.	High (~10^7^ particles/mL)	Medium (single sample)	~30–60 min
Flow Cytometry	Detects EVs using light scattering and fluorescent markers.	Moderate (~10^8^ particles/mL)	Medium to High	~1–2 h
Western Blotting	Identifies specific EV proteins via antibody detection.	High for target proteins	Low (single sample)	~4–6 h
Transmission Electron Microscopy (TEM)	Visualizes EV morphology and size at nanoscale resolution.	High (single EV detection)	Very Low	~1–2 days
Dynamic Light Scattering (DLS)	Measures EV size distribution based on scattering of light.	Moderate (~10^8^ particles/mL)	Medium	~30–60 min
Mass Spectrometry	Profiles EV proteins, lipids, or metabolites for in-depth molecular analysis.	High for molecular detail	Low to Medium	~1–2 days
Surface Plasmon Resonance (SPR)	Measures binding interactions between EV components and specific ligands.	High for target molecules	Low to Medium	~2–4 h
Enzyme-Linked Immunosorbent Assay (ELISA)	Quantifies specific EV markers using antigen-antibody interactions.	High (picogram levels)	Low to Medium	~4–6 h
qPCR for EV RNA Analysis	Quantifies RNA content in EVs through reverse transcription and amplification.	High for RNA detection	Low to Medium	~3–6 h

**Table 5 ijms-26-00189-t005:** Summarizing the use of EVs for disease diagnosis and monitoring, with their advantages and disadvantages in both clinical and research applications.

Disease Diagnosis and Monitoring	Advantages	Disadvantages
Cancer Disease	EVs carry tumor-specific markers for early detection and monitoring of cancer	Lack of standardized methods for isolating and analyzing EVs makes reproducibility challenging
	Non-invasive collection from biofluids reduces the need for biopsies	Heterogeneity of EVs from different sources may complicate interpretation of diagnostic results
	EVs reflect tumor heterogeneity, providing a broader understanding of the disease than traditional biopsies	EV analysis can be time-consuming and require advanced techniques such as next-gene sequencing or mass spectrometry
Cardiovascular Disease	Elevated EVs levels are associated with heart failure, hypertension, and atherosclerosis, aiding early detection	Variability in EV concentrations and composition based on patient conditions can affect diagnostic accuracy
	EVs can reflect endothelial damage and inflammation, offering insights into disease progression	Requires highly sensitive and specific techniques, making clinical application costly
Neurodegenerative Disease	EVs can cross the blood-brain barrier, providing insight into brain conditions from peripheral biofluids	Difficulty in isolating brain-specific EVs from biofluids
	EVs carry neurodegenerative markers aiding early diagnosis of diseases like Alzheimer’s and Parkinson’s	Lack of sensitivity in detecting low-abundance biomarkers in early disease stages
Autoimmune Disease	EVs can carry autoantigens and inflammatory markers relevant to diseases like rheumatoid arthritis and lupus	The diversity of EV cargo complicates interpretation in autoimmune disease contexts
	EV analysis can aid in determining disease activity and progression	Requires large-scale validation in diverse populations for clinical applications
Metabolic Diseases	EVs carry miRNAs and proteins linked to insulin resistance and glucose metabolism, allowing for early diagnosis of T2D and NAFLD	EV biomarkers may not be disease-specific and can overlap with other inflammatory conditions
	Non-invasive sampling for disease monitoring (blood or urine)	Need for standardized assays to interpret the role of EVs in disease progression across populations

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
