# Peer review of "Extracellular Vesicles: Advanced Tools for Disease Diagnosis, Monitoring, and Therapies"

_ijms, 2024, doi:10.3390/ijms26010189_

Round 1

Reviewer 1 Report (New Reviewer)

Comments and Suggestions for Authors

The review presented is interesting. However, there are some comments raised for your consideration:

1. While various methods are mentioned, the authors should provide a more in-depth discussion and comparison of EV isolation techniques. Here are some suggestions for expanding this section:

A. The principles underlying each isolation technique should be explained.

B. Highlight and review specific commercially available kits or established systems that are representative of each principle. Compare and discuss their performance, advantages, and limitations in terms of yield, purity, and reproducibility.

C. Include information regarding the challenges of EV isolation from different types of biological fluids.

D. Microfluidic-based isolation for EV is a broad field. A review of the principles incorporated into microfluidic systems, such as size-based separation and immunocapture, would be informative for this section.

2. The author should include a representative illustration for microfluidic isolation method for Figure 3.

3. The authors should revise Table 3. The “Principle” and “Applications” columns provide overlapping information. For the “Principle” column, details such as scattered light, electron beam, and Brownian motion should be included. Furthermore, the authors should consider providing additional information, such as sensitivity, throughput, and processing time.

4. The addition of isolation methods in Table 3 makes the table inconsistent. The authors should create a separate table specifically for isolation methods.

Author Response

The review presented is interesting. However, there are some comments raised for your consideration:

  1. While various methods are mentioned, the authors should provide a more in-depth discussion and comparison of EV isolation techniques. Here are some suggestions for expanding this section:
  2. The principles underlying each isolation technique should be explained.

Thank you for your valuable suggestion. In response to your comment, we have incorporated detailed explanations of the principles underlying each isolation technique used in our study. Text in blue from 450 line to 572 line

  1. Highlight and review specific commercially available kits or established systems that are representative of each principle. Compare and discuss their performance, advantages, and limitations in terms of yield, purity, and reproducibility.

Thank you for your thoughtful suggestion. In response, we have incorporated a detailed discussion in the revised manuscript that highlights and reviews specific commercially available kits established systems representative of each principle. Text in blue from 543 line to 559 line

  1. Include information regarding the challenges of EV isolation from different types of biological fluids.

We appreciate the reviewer’s suggestion to address the challenges associated with extracellular vesicle isolation from different biological fluids. This has been incorporated into the revised manuscript now highlights that the complexity and variability of biological fluids, such as plasma, urine, cerebrospinal fluid and others present significant challenges for EV isolation. We have included key factors as high abundance of contaminants, low EV concentrations, variable viscosity and composition and different isolation techniques exhibit varying efficiencies, yields, and purities across fluids due to differences in particle composition and matrix effects. Text in blue from 498 line to 519 line

  1. Microfluidic-based isolation for EV is a broad field. A review of the principles incorporated into microfluidic systems, such as size-based separation and immunocapture, would be informative for this section.

We thank the reviewer for the valuable suggestion to include a review of the principles incorporated into microfluidic systems for extracellular vesicle. This has been addressed in the revised manuscript. Specifically, we now provide an overview of the fundamental principles applied in microfluidic-based EV isolation, emphasizing size-based separation, immunocapture, acoustic electrophoretic techniques and aspects of integration and automation facilitating rapid and user-friendly workflows. . Text in blue from 5038 line to 519 line

  1. The author should include a representative illustration for microfluidic isolation method for Figure 3.

We thank the reviewer for the suggestion to include a representative illustration of the microfluidic isolation method for Figure 3. This has been addressed in the revised manuscript.

  1. The authors should revise Table 3. The “Principle” and “Applications” columns provide overlapping information. For the “Principle” column, details such as scattered light, electron beam, and Brownian motion should be included. Furthermore, the authors should consider providing additional information, such as sensitivity, throughput, and processing time.

We appreciate the reviewer’s insightful comments regarding Table 3 and have made the recommended revisions. Principle and Applications columns have been refined to eliminate redundancy. The “Principle” column now focuses on the fundamental mechanisms underlying each method (e.g., scattered light, electron beam interaction, or Brownian motion), while the “Applications” column has been eliminated. New columns have been added to the table to provide information on advantages, limitations, and processing time for each technique. These additions offer readers a more comprehensive understanding of the strengths and limitations of each method

  1. The addition of isolation methods in Table 3 makes the table inconsistent. The authors should create a separate table specifically for isolation methods.

We appreciate the reviewer’s suggestion to address the inconsistency caused by the addition of isolation methods in Table 3. This has been resolved in the revised manuscript by separating the information into two distinct tables: Updated Table 3: This table now exclusively focuses on characterization methods for extracellular vesicles, detailing the principle (e.g., scattered light, electron beam, Brownian motion), advantages, limitations, and processing time for each technique. New Table: A separate table has been created specifically for EV isolation methods. This table provides a comprehensive summary of various isolation techniques, categorized by their principles (e.g., ultracentrifugation, size-exclusion chromatography, immunoaffinity capture, precipitation, microfluidics), and includes details on their principles advantages, limitations and processing time.

Reviewer 2 Report (New Reviewer)

Comments and Suggestions for Authors

In this review, the authors provide an updated analysis of the significance of EVs, emphasizing their mechanisms of action and applications in diagnosing and treating various diseases. Additionally, the review examines the existing limitations and future potential of EVs, offering practical recommendations to address current challenges and enhance their clinical viability. While this review is a comprehensive contribution to the EV field, certain sections lack depth and require further discussion. Thus, a major revision is recommended.

Specific comments are as follows:

1.     Figure 2: The authors list various EV subtypes but focus solely on the biogenesis of exosomes and microvesicles. The inclusion of other subtypes should be considered.

2.     Common Isolation Methods: This section is underdeveloped. The authors should elaborate on the principles, advantages, and limitations of commonly used isolation methods. Additionally, challenges such as scalability, reproducibility, and contamination risks should be discussed in detail.

3.     Therapeutic Administration of EVs: A detailed discussion on the administration strategies for EVs as therapeutic agents is needed.

4.     Lines 492–493: The references to MIFlowCyt and MISEV frameworks for EV characterization are vague. These guidelines should be described in more detail, specifying their requirements and significance in standardizing EV characterization

5.     Loading Techniques (Section 7): The authors briefly mention the lack of efficient loading techniques but do not provide sufficient detail. This section should be expanded to include a categorized discussion of current loading strategies, their limitations, and recent advancements. Topics such as contamination and the loading of large cargoes (e.g., exosomal mRNAs) should be highlighted, with reference of studies from the past two years.

6.     A summary of recent clinical cases and the current status of EVs in diagnostic and therapeutic applications is needed. Additionally, challenges related to EV bioengineering, including purification, storage, and regulatory compliance, should be addressed. The authors are encouraged to discuss recent references to support the topics, such as 10.1016/j.tibtech.2024.08.007ï¼› doi.org/10.1038/s41565-021-00931-2, among others focusing on clinical translation.

Author Response

Specific comments are as follows:

  1. Figure 2: The authors list various EV subtypes but focus solely on the biogenesis of exosomes and microvesicles. The inclusion of other subtypes should be considered.

We thank the reviewer for highlighting the need to include other EV subtypes in Figure 2. This has been addressed in the revised manuscript. Figure 2 has been updated to provide a more comprehensive depiction of EV subtypes

  1. Common Isolation Methods: This section is underdeveloped. The authors should elaborate on the principles, advantages, and limitations of commonly used isolation methods. Additionally, challenges such as scalability, reproducibility, and contamination risks should be discussed in detail.

We thank the reviewer for pointing out the need for further elaboration on the isolation methods. The section on "Common Isolation Methods" has been significantly expanded in the revised manuscript to address the reviewer’s concerns. Text in blue from 450 line to 571 line

  1. Therapeutic Administration of EVs: A detailed discussion on the administration strategies for EVs as therapeutic agents is needed.

We thank the reviewer for highlighting the need to expand the discussion on the administration strategies for EVs as therapeutic agents. This has been addressed in the revised manuscript by including a detailed and comprehensive discussion of commonly used administration routes for EV-based therapies and functionalization of EV surfaces with targeting ligands (e.g., peptides, antibodies) to enhance homing to specific tissues. Text in blue from 919 line to 957 line

  1. Lines 492–493: The references to MIFlowCyt and MISEV frameworks for EV characterization are vague. These guidelines should be described in more detail, specifying their requirements and significance in standardizing EV characterization

We thank the reviewer for emphasizing the need to elaborate on the MIFlowCyt and MISEV frameworks. The revised manuscript now provides a more detailed description of these guidelines. Text in blue from 659 line to 683 line

  1. Loading Techniques (Section 7): The authors briefly mention the lack of efficient loading techniques but do not provide sufficient detail. This section should be expanded to include a categorized discussion of current loading strategies, their limitations, and recent advancements. Topics such as contamination and the loading of large cargoes (e.g., exosomal mRNAs) should be highlighted, with reference of studies from the past two years. 

Thank you for your valuable feedback. We appreciate the suggestion to expand Section 7 on Loading Techniques. In response, we have revised and expanded this section to provide a more detailed and categorized discussion of current loading strategies, including highlighted key challenges such as contamination during loading and strategies for loading large cargoes. Text in blue from 1085 line to 1128 line

  1. A summary of recent clinical cases and the current status of EVs in diagnostic and therapeutic applications is needed. Additionally, challenges related to EV bioengineering, including purification, storage, and regulatory compliance, should be addressed. The authors are encouraged to discuss recent references to support the topics, such as 10.1016/j.tibtech.2024.08.007ï¼› doi.org/10.1038/s41565-021-00931-2, among others focusing on clinical translation.

Thank you for your insightful feedback. In response to your suggestion, we have thoroughly revised the manuscript to incorporate a detailed summary of recent clinical cases and the current status of extracellular vesicles (EVs) in diagnostic and therapeutic applications. we have incorporated recent studies and references, including those you mentioned: DOI: 10.1016/j.tibtech.2024.08.007: We have cited this recent review, which discusses the evolving landscape of EV-based therapeutics and diagnostic strategies. DOI: 10.1038/s41565-021-00931-2: We have referenced this study to highlight advancements in EV bioengineering, particularly regarding the optimization of EV production and functionalization for clinical applications. Text in blue from 1016 line to 1032 line

Round 2

Reviewer 1 Report (New Reviewer)

Comments and Suggestions for Authors

Minor suggestions:

1. Ensure consistency in numerical expressions (e.g., 10^6 or 106)

2. Ensure consistency in icons, fonts, and labeling in figures.

Reviewer 2 Report (New Reviewer)

Comments and Suggestions for Authors

The authors have satisfactorily addressed all my concerns. Well done!

This manuscript is a resubmission of an earlier submission. The following is a list of the peer review reports and author responses from that submission.

Round 1

Reviewer 1 Report

Comments and Suggestions for Authors

1.     What is the main question addressed by the research?

Pedro Lorite and co-collaborators in this review describe the involvement of EVs in diseases and what they can do for diagnosis, prognosis, and therapy.

2.     Do you consider the topic original or relevant in the field? Does it address a specific gap in the field?

It is a very relevant topic; however, it seems to be a very general review trying to cover very broad topics.

3. What does it add to the subject area compared with other published material?

Previous reviews appear to touch on these topics. However, I believe focusing on the mechanisms of action of vesicles would enhance the manuscript.

4. Are the conclusions consistent with the evidence and arguments presented, and do they address the central question posed?

The authors are encouraged to reduce the conclusions.

5. Are the references appropriate?

Yes, they are appropriate.

6. Please include any additional comments on the tables and figures.

1 Authors are advised to read the manuscript carefully for inconsistencies and to abbreviate words in the first appearance.

2 Authors are encouraged to include the review's objective in the abstract.

3 Authors are encouraged to include references to the isolation methods.

4 I am not entirely convinced by point 4 since it speaks very generally of isolation methods, especially point 4.4, which deals with something other than protein and NA isolation. Lines 288-297 seem to me to be in points 4.2.….

5 It is recommended to the authors to give a more precise approach to section 5.1.1 as they did in 5.1.2, which put mechanisms of action of EVS, i.e. (cargo) and participation, how it is used in diagnosis. Since this is very general, it should talk about proteins that have been proposed in the diagnosis of diseases,

6 Point 5.2 seems to me to be very general in the first four paragraphs (lines 462-486); these points in these paragraphs feel repetitive; since these points were already discussed above, it gets interesting from line 487, which antiviral agents, give examples if they have worked in cell lines, models, etc.

It is very repetitive with exosomes and microvesicles.

For the case of 5.3, it is also recommended to put pathways of how it is proposed.

7 In point 6, lines 593-613, it seems the disadvantages of isolation methods (already discussed in another section), not Challenges and Future. Section 7 looks like section 6. It is recommended to complement them.

Author Response

1 Authors are advised to read the manuscript carefully for inconsistencies and to abbreviate words in the first appearance.

 Thank you for the reminder. We will carefully review the manuscript to ensure consistency throughout and make sure that all abbreviations are introduced at their first appearance as suggested.

2 Authors are encouraged to include the review's objective in the abstract.

 Thank you for your suggestion. We will revise the abstract to clearly include the review's objective (appear in blue), ensuring it aligns with the overall focus and purpose of the manuscript.

3 Authors are encouraged to include references to the isolation methods.

Thank you for your suggestion. We will ensure that references to the isolation methods used for extracellular vesicles (EVs) are included, providing a more detailed and scientifically robust explanation of the techniques applied. This will enhance the clarity and reproducibility of the methodology described in the manuscript.

 4 I am not entirely convinced by point 4 since it speaks very generally of isolation methods, especially point 4.4, which deals with something other than protein and NA isolation. Lines 288-297 seem to me to be in points 4.2

 Thank you for your feedback. I understand your concerns about point 4, especially regarding its general approach to isolation methods. We make revision to clarify the specific techniques, to improve the manuscript's chapter structure and coherence.

5 It is recommended to the authors to give a more precise approach to section 5.1.1 as they did in 5.1.2, which put mechanisms of action of EVS, i.e. (cargo) and participation, how it is used in diagnosis. Since this is very general, it should talk about proteins that have been proposed in the diagnosis of diseases,

 Thank you for your valuable feedback. We agree that section 5.1.1 (now 6.1.1) would benefit from a more precise approach, similar to what was done in 5.1.2. (now 6.1.2). In the revised version, we included specific examples of proteins that are being investigated for their diagnostic potential in various diseases.

6 Point 5.2 seems to me to be very general in the first four paragraphs (lines 462-486); these points in these paragraphs feel repetitive; since these points were already discussed above, it gets interesting from line 487, which antiviral agents, give examples if they have worked in cell lines, models, etc.

Thank you for your constructive feedback. We acknowledge that the first four paragraphs of point 5.2 (now 6.2) may appear repetitive, and we have revised this section to reduce redundancy and ensure a more concise presentation.

It is very repetitive with exosomes and microvesicles.

Thank you for your feedback. We acknowledge that certain sections of the manuscript may seem repetitive when discussing exosomes and microvesicles. We reviewed these parts ensuring that each section adds unique insights without redundant information.

For the case of 5.3, it is also recommended to put pathways of how it is proposed.

 Thank you for the suggestion. We will refine section 5.3(now 6.3) to include more specific pathways, detailing how these are proposed in the context of EVs.

7 In point 6, lines 593-613, it seems the disadvantages of isolation methods (already discussed in another section), not Challenges and Future. Section 7 looks like section 6. It is recommended to complement them.

Thank you for your feedback. We have been implemented to avoid repetitions and appear in a more orderly manner in the text.

Reviewer 2 Report

Comments and Suggestions for Authors

The manuscript aims to provide the current knowledge about extracellular vesicles and discuss their potential in disease diagnosis and treatment. The review is well-written and provides a comprehensive overview of the topic. The figures and tables are informative. However, the manuscript is a bit lengthy and could be more concise. The key points are not very clear and some of the sections could be better organized. There are a few points I think that could be considered to further strengthen the paper.

1.     The title is a little bit mouthful and not closely relevant to the major content of this manuscript. I think it should not mention the tools of EV research, like characterization and enrichment, but emphasis the application of EVs in disease diagnosis and treatment.

2.     The size of apoptotic bodies in Figure 1 is not consistent with the description in line 101.

3.     Please confirm the marker of exosomes is HSC70, HSP70, or both in second row of Table 1.

4.     Several common functions listed in line 224 to 246 could be combined since they are partly overlapped, for example, i and iii, ii and vi, vii and viii.

5.     It would be beneficial to reorganize Table 2 since the information in the middle column is a little bit hard to read.

6.     Section 4 could be considered to include a table to list the advantages and limitations of each isolation method more clearly. The minimal information for studies of extracellular vesicles published by ISEV this year could be considered to reference here.

Welsh, Joshua A., et al. "Minimal information for studies of extracellular vesicles (MISEV2023): From basic to advanced approaches." Journal of extracellular vesicles 13.2 (2024): e12404.

7.     Table 3, 4, and 5 are also partly overlapping with the description in each section. I would recommend simplifying Table 3 and 4 by removing redundant information, and replacing Table 5 with helpful figures.

Comments on the Quality of English Language

There are several typos in the manuscript. For example, capitalization problem in line 60, line 459, first blank of Table 3. Redundant abbreviation in line 314, and so on. Please check the whole manuscript.

Author Response

  1. The title is a little bit mouthful and not closely relevant to the major content of this manuscript. I think it should not mention the tools of EV research, like characterization and enrichment, but emphasis the application of EVs in disease diagnosis and treatment.

Thank you very much for your valuable feedback. We agree. The title has been corrected to focus on the role of diagnosis, monitoring and therapy.

  1. The size of apoptotic bodies in Figure 1 is not consistent with the description in line 101.

Thank you for your observation. I am sorry fot the mistake. We have corrected

  1. Please confirm the marker of exosomes is HSC70, HSP70, or both in second row of Table 1.

Thank you for your observation. We have confirmed HSP70 as exosome marker

  1. Several common functions listed in line 224 to 246 could be combined since they are partly overlapped, for example, i and iii, ii and vi, vii and viii.

Thank you very much for your valuable feedback. We agree. These common functions have been rewritten by combining some of them to avoid overlaps. (appear text in blue)

  1. It would be beneficial to reorganize Table 2 since the information in the middle column is a little bit hard to read.

Thank you for your suggestion. We agree, The table 2 has been reorganized for easier reading.

  1. Section 4 could be considered to include a table to list the advantages and limitations of each isolation method more clearly. The minimal information for studies of extracellular vesicles published by ISEV this year could be considered to reference here. Welsh, Joshua A., et al. "Minimal information for studies of extracellular vesicles (MISEV2023): From basic to advanced approaches." Journal of extracellular vesicles 13.2 (2024): e12404.

Thank you very much for your valuable feedback. I have included in text form The 2023 International Society for Extracellular Vesicles (ISEV) guideline that emphasizes the importance of standardizing the terminology used for EVs (from line 197 to 208)

  1. Table 3, 4, and 5 are also partly overlapping with the description in each section. I would recommend simplifying Table 3 and 4 by removing redundant information, and replacing Table 5 with helpful figures.

Thank you very much for your valuable feedback. We agree. We have removed tables 4 and 5, and have included a new figure (figure 4) in Challenges and future directions in extracellular vesicles research chapter

Round 2

Reviewer 1 Report

Comments and Suggestions for Authors

The authors addressed all issues raised by this reviewer.

Author Response

Comment: The authors addressed all issues raised by this reviewer.

Response: Thank you very much for your positive feedback and for taking the time to review our manuscript thoroughly. We appreciate your comments and are glad that the revisions have addressed all the issues you raised. Your insights have been invaluable in strengthening the clarity and impact of our work. Thank you again for your constructive feedback and support.

Reviewer 2 Report

Comments and Suggestions for Authors

I appreciate the authors for responding to my previous comments. However, there are still a few problems that need to be checked for further processing of this manuscript.

1.       Identifying microvesicles can be challenging due to their heterogeneity and overlap with other EVs. As far as I know tetraspanins and flotillins are often enriched in MVs. So, it would be beneficial to double-check the markers described in line 88-93 and listed in Table 1.

2.       The size of apoptotic bodies in Table 1 is not consistent with the previous description. Please check the whole manuscript for any typo problems.

3.       The novelty of this paper is still not very clear since there are so many similar reviews that have been published talking about in most of the sections of this paper. The most important information should be highlighted to improve the impact of it.

Comments on the Quality of English Language

The whole manuscript needs to be reorganized since there are many issues about the format even though this is not the most important problems at this stage.

Author Response

I appreciate the authors for responding to my previous comments. However, there are still a few problems that need to be checked for further processing of this manuscript.

Comment 1: Identifying microvesicles can be challenging due to their heterogeneity and overlap with other EVs. As far as I know tetraspanins and flotillins are often enriched in MVs. So, it would be beneficial to double-check the markers described in line 88-93 and listed in Table 1.

Response 1: Thank you for this valuable feedback. We agree that the heterogeneity of microvesicles (MVs) and their overlap with other extracellular vesicles (EVs) make specific identification challenging. Tetraspanins and flotillins are indeed markers commonly associated with MVs, as you mentioned. We will review the markers listed in lines 88–93 and Table 1 to ensure accuracy and consistency, taking into account the enrichment patterns of tetraspanins,and flotillins. This adjustment will enhance clarity and precision in EV types. Thank you again for highlighting this point.

Comment 2: The size of apoptotic bodies in Table 1 is not consistent with the previous description. Please check the whole manuscript for any typo problems.

Response 2: Thank you for pointing out this discrepancy. We will carefully review Table 1 and the entire manuscript to ensure that the size range of apoptotic bodies is consistent with our initial description and correct any typographical errors. Maintaining accuracy in these details is essential for clarity, and we appreciate your attention to this matter.

Comment 3: The novelty of this paper is still not very clear since there are so many similar reviews that have been published talking about in most of the sections of this paper. The most important information should be highlighted to improve the impact of it.

Response 3: Thank you for this valuable feedback. We understand the importance of clearly highlighting the unique contributions of our review. In response, we will revise the manuscript to emphasize the novel aspects, particularly our focus on the integration of organ-on-chip (OoC) platforms and artificial intelligence (AI) with extracellular vesicle (EV) research. These elements provide a fresh perspective on how EVs can be optimized for diagnostics and therapeutic applications by reducing experimental bias and enhancing research consistency. We will also underscore the distinct sections on the standardization of EV isolation, characterization, and quantification, as well as recent advancements in overcoming EV heterogeneity and loading efficiency—topics that address current challenges in the field. (All these news aspects incorporated, such as new text, new table and new reference,s appear in the text in red)

We aim to distinguish this review from existing literature and clarify the impact of our insights on future EV research directions. This review contributes to the field by integrating recent discoveries, promoting methodological rigor, and suggesting specific strategies for translating EV research into clinical practice—particularly in personalized medicine and precision therapies. The review identifies key areas for future investigation that may further enhance EV viability for clinical use, helping bridge the gap between EV research and clinical application.

Thank you again for your insightful suggestions to improve the manuscript’s clarity and relevance.

Round 3

Reviewer 2 Report

Comments and Suggestions for Authors

I appreciate the authors for responding to my previous comments. However, I still hard to find the novelty of this paper since most of the content have been described in many previous publications. Besides, there are still many format issues even though I have mentioned in the previous two review processes. Like but not limited to typos in the title in line 441. The numbering format of section 2 and 4 are not consistent with section 5 and 6. Please check the whole manuscript seriously and carefully since it would be helpful for readers to focus on the content but not these confusing issues.

Comments on the Quality of English Language

Please see comments and suggestions.

Author Response

Reviewer 2. Round 3

We submit our manuscript to IJMS after making revisions and providing new data. However, we have made modifications to improve the manuscript. In Chapter 4, Integrative Proteo-Transcriptomic Analyses has been added and subsequently developed in Subchapter 4.2 (highlighted in blue in the text). Additionally, Chapter 7 has been expanded to include aspects of Food-Derived EVs (highlighted in blue in the text). Also, we have added new references

We appreciate the reviewer’s comments. However, we respectfully disagree with the assertion that the manuscript lacks novelty and that most of its content has already been described in previous publications. This review is, to date, the most comprehensive in integrating all aspects related to extracellular vesicles and their applications in diagnosis, monitoring, and therapy, including recent advancements in technology, multi-omics approaches, emerging clinical applications, and future perspectives. While many reviews exist, this one stands out by integrating the latest findings in EV research and their implications for real-world applications. It synthesizes data from diverse fields to provide a holistic and current analysis, setting the stage for future advancements in EV-based science and medicine. By addressing these unique aspects, the review serves as both a foundational resource for newcomers to the field and a strategic guide for researchers aiming to push the boundaries of EV applications in medicine. Additionally, we highlight underexplored aspects of EV research, such as their role in multi-omics integration and the potential of AI-driven approaches to EV analysis, which have not been extensively covered in previous publications. To better convey this novelty, we have included interesting aspects of food-derived EVs and their posible medical in implications with unique contributions and position our review within the existing literature. We reiterate that this review offers significant added value by consolidating scattered information from multiple studies and presenting an updated framework that can serve as a guide for researchers and clinicians interested in the topic. It not only provides a broad overview but also includes a critical analysis of current limitations and future directions, which substantially differentiates it from previous reviews.

While the reviewer indicates that the content has been previously addressed in other reviews, no evidence or specific examples have been provided to suggest how this review could be further improved or what aspects they consider novel. Additionally, no specific works are identified that purportedly cover the same breadth of topics or level of integration as this manuscript. We believe this observation does not adequately reflect a thorough analysis of the scope and unique contributions of our work. It would have been greatly helpful to receive more specific suggestions or comments from the reviewer on how to improve the approach or include additional relevant elements if deemed necessary. In this regard, we have had to refine and strengthen our review based on our own judgment, as no feedback has been provided to guide such improvements.

We do not consider it appropriate to reject the publication of a review solely based on a subjective opinion from the reviewer, particularly if no specific changes or modifications are suggested to improve the manuscript. Limiting the evaluation to observations related to formatting, without providing concrete suggestions regarding the content or its approach, hinders constructive feedback and the advancement of the work.

Format Issues:

We apologize for the oversight regarding formatting inconsistencies and typographical errors, including the title in line 441 and the numbering format in sections 2, 4, 5, and 6. These have been carefully reviewed and corrected throughout the manuscript. We fully understand your concern about how formatting issues might distract readers from the content of the manuscript. By addressing these issues rigorously, we aim to improve the overall readability and ensure that the focus remains on the scientific content rather than technical distractions.
